# NRGPT: An Energy-based Alternative for GPT

**Nima Dehmamy**[*]
IBM Research

**Benjamin Hoover**[*]
IBM Research; Georgia Tech

**Bishwajit Saha**[*]
IBM Research

**Leo Kozachkov**
Brown University

**Jean-Jacques Slotine**
MIT

**Dmitry Krotov**[*]
IBM Research

## ABSTRACT

Generative Pre-trained Transformer (GPT) architectures are the most popular design for language modeling. Energy-based modeling is a different paradigm that views inference as a dynamical process operating on an energy landscape. We propose a minimal modification of the GPT setting to unify it with the EBM framework. The inference step of our model, which we call eNeRgy-GPT (NRGPT), is conceptualized as an exploration of the tokens on the energy landscape. We prove, and verify empirically, that under certain circumstances this exploration becomes gradient descent, although they don't necessarily lead to the best performing models. We demonstrate that our model performs well for simple language (Shakespeare dataset), algebraic ListOPS tasks, and richer settings such as OpenWebText language modeling. We also observe that our models may be more resistant to overfitting, doing so only during very long training.

Transformers represent a dominant paradigm in autoregressive language modeling (Vaswani et al., 2017). In a typical setting, a sequence of tokens describing a text is passed through several transformer layers and mapped onto a new sequence, which is a copy of the original one shifted by one token and appended by the token that follows the initial sequence. At training time, this network is trained through self-supervised training, and at inference time the network is used for next token prediction. This is the standard Generative Pre-trained Transformer (GPT) setting, which is the first step in Large Language Model (LLM) design (Radford et al., 2018).

Energy-based modeling (LeCun et al., 2006) is another prominent paradigm in modern AI landscape that historically goes back to Hopfield Networks (Hopfield, 1982). In this framework the operation of the neural network is defined by a scalar energy function. Proper samples generated by the model (those that resemble training data) correspond to low energy states, while unrealistic samples (with large deviations from the training data distribution) correspond to high energy states. Thus EBMs are particularly appealing for their theoretical properties, because they view the "forward pass" through a deep network as an explicit optimization problem that maximizes the likelihood of the input data. Practically, this approach unlocks better tooling for systematic exploration of the solution space and enables natural solutions for both variable computation and model alignment via regularizers (see our extended discussion on EBMs for LLMs in Appendix C).

Although at face value these two approaches look very different, in recent years a growing number of studies hint at deep connections. Von Oswald et al. (2023) showed evidence that in-context learning (ICL) may be gradient descent by constructed an explicit weights such that the forward pass was GD on MSE loss. Ahn et al. (2024) further showed that transformers learn a preconditioned GD for ICL. However, both of these works make significant simplifications, such as considering only *linear* transformers, omitting the softmax.

Other works have have attempted to reconcile transformers and EBM from several angles. For instance, the Energy Transformer (Hoover et al., 2023) is an architecture, which is simultaneously a transformer and an energy-based model. In the image domain, the typical setting would be to reconstruct a set of masked tokens (patches) given the set of open tokens. The network solves this task by performing a gradient descent of the energy on the space of tokens at inference time. This architecture is inspired by associative memory models (Krotov and Hopfield, 2016) and for this reason solves the following problem: given a partially incomplete pattern – complete it in a meaningful way.

---

[*]Equal contribution.

This aspect of the core design makes it difficult to apply Energy Transformers to GPT settings, in which the sequence needs to be transformed to a shifted sequence by means of going through the network. Intuitively, in Energy Transformers the masked tokens need to evolve rapidly to match the missing parts of the pattern (e.g., image or graph), while the open tokens need to stay almost constant to barely adjust for the smooth transitions between the masked and the open tokens within the pattern. This is in drastic contrast with the GPT setting, in which there are no masked tokens at all. Rather, every token needs to evolve into the following token in the sequence.

A different line of work is inspired by "System 2" thinking and attempts to design an energy-based network for processing language (Gladstone et al., 2025). In this study, transformers are used as an architectural motif that casts text into a scalar energy function. While models of this nature have benefits for language processing, they belong exclusively to the class of energy-based models, and are unrelated to the GPT settings, commonly used in most LLMs.

Individual modules within the transformer block, such as attention, have also been studied from the perspective of inference time optimization (Geshkovski et al., 2023; 2024). In this line of work, peculiar clustering properties of tokens have been observed. Energy-based optimization has also been studied in (Yang et al., 2022) from the perspective of majorization-minimization algorithms.

Despite this growing list of studies dedicated to synergies between autoregressive transformers and energy-based models, at present it remains unknown how to cast the commonly used GPT setting into a well-defined energy-based framework. Our work tackles this gap. We refer to our model as e**N**e**R**gy **G**enerative **P**re-trained **T**ransformer or NRGPT. The input sequence of tokens is mapped onto a shifted sequence of tokens, which includes the next word, see Figure 1. The mapping is performed by a neural network, which recurrently applies the NRGPT block to the sequence of tokens. Each application of the block uses gradients of the network energy functions to update the state of the tokens. Each token has its own energy landscape, which is dependent on the states of other tokens. Specifically, our contributions are:

- We design an energy function and an update rule that describes the GPT setting with several possible variants including learnable inference rate and normalization operations: LayerNorm and RMSNorm.
- We obtain excellent results on nested ListOPS tasks, including arithmetic operations, min/max selection, etc.
- We show the feasibility of using NRGPT for language modeling on Shakespeare and OpenWebText datasets.
- We do a systematic comparison of performance scaling of recurrent transformers and NRGPT.
- We study empirically the properties of dynamical trajectories of tokens on the energy landscapes of our models.

## 1  ENERGY-BASED MODELING

In generative modeling our goal is to generate samples with a distribution close to observed datapoints. If we manage to learn an approximate likelihood function for the dataset, we can generate data by sampling. This is also the premise Energy-Based Models (EBM). An example of EBM would be Dense Associative Memory (Krotov and Hopfield, 2016), where datapoints are stored in minima of an energy function. But more generally, the energy can represent a negative-log-likelihood, $E(\mathbf{x}) = -\log P(\mathbf{x})$. In this case, the global minima of the energy represent maximum likelihood solutions. The deeper the energy, the higher the likelihood of that datapoint. One strategy to train an EBM is to first learn the energy function by fitting the distribution of the data. The sampling process would then be separate from learning the energy.

However, in high dimensional data, learning the distributions is notoriously difficult due to the curse of dimensionality. Diffusion models solve this problem by starting from high noise and cooling down. Diffusion models do not learn an explicit energy function, only its gradients, the score function. Yet, having an explicit energy function could enable us to explore the solution space in ways not easily afforded by the implicit score function of diffusion models. So can we build a model which learns the energy directly?

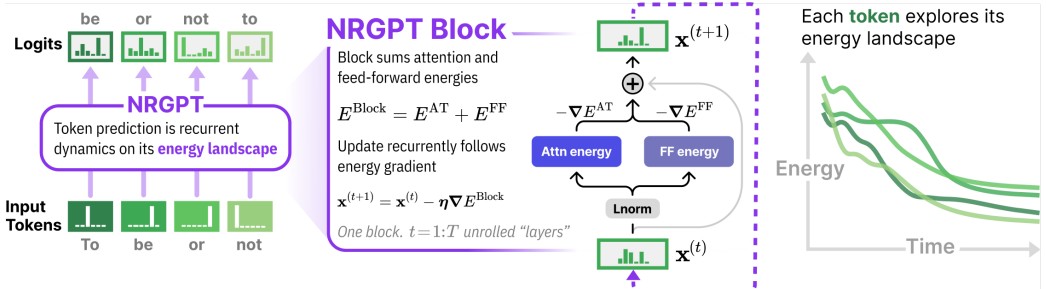

Figure 1: **NRGPT** casts the standard GPT setting into an energy-based framework. The network is defined as the sum of two energies: an **attention** energy and a **feedforward** energy. Each **token** is transformed into the next token by exploring the energy landscape. Recurrent application of the NRGPT block produces a dynamical system where each token can be thought of as a particle moving on the network's energy landscape.

Similar to diffusion models, we use an end-to-end process, where learning the energy and generating datapoints are all done in one pass. The key idea is to have a differentiable sampling process which allows us to learn the parameters of the energy during sampling. Since real datapoints should have low energies, we choose a gradient-based sampling process. Note that we do not need to descend all the way to a minimum (i.e. maximum likelihood solution), since we want diverse samples. Instead, we do a fixed number of energy GD steps and demand that the final point matches real datapoints.

$$\text{Generated data: } \mathbf{x}^{(T)}, \qquad \mathbf{x}^{(t+1)} = \mathbf{x}^{(t)} - \boldsymbol{\eta}^{(t)}\boldsymbol{\nabla}E(x^{(t)}) \tag{1}$$

for a fixed number of steps $T$, where $\mathbf{x}^{(0)}$ is some random initial point. Here $\boldsymbol{\eta}^{(t)}$ is a matrix that may depend on $\mathbf{x}$. This matrix has many different names, e.g., kinetic rates in physics, preconditioner in optimization, etc. We will call this matrix the *inference rate*, since it determines the size of the steps that the inference dynamics takes on the energy landscape. But how do we judge whether the output matches a real datapoint? One way would be to have a judge, like the discriminator in a GAN. Another setting where judging the output is more natural is autoregressive language modeling where the new datapoints are the next tokens and can be matched to the training text. In this case $\mathbf{x} \in \mathbb{R}^{N \times D}$ represents a real data sequence of length $N$ embedded in $D$ dimensions. In causal language modeling the energy should take $\mathbf{x}_{<N} = (\mathbf{x}_1 \dots \mathbf{x}_{N-1})$ as input and predict $\mathbf{x}_N$, as in

$$\mathbf{x}_N^{(t+1)} = \mathbf{x}_N^{(t)} - \boldsymbol{\eta}\boldsymbol{\nabla}E(\mathbf{x}_N^{(t)}|\mathbf{x}_{<N}^{(t)}) \tag{2}$$

Following the observations of the Energy Transformer (ET), we will show that one can choose a parametrization for $E$ such that the process of $T$-step GD closely resembles the forward-pass through a $T$ layer GPT transformer with a weight-sharing pattern.

## 2 NRGPT MODULE

$$E_A = -\frac{1}{\beta}\sum_{h=1}^{H}\log\left(\sum_{B<A}\exp\left(\beta\,\mathbf{g}_B^T\boldsymbol{J}^h\mathbf{g}_A\right)\right) - \sum_{B=1}^{N}\mathbf{g}_B^T\mathbf{W_2}\sigma\left(\mathbf{W_1}\mathbf{g}_B\right) \tag{3}$$

In this section we will start from the structure of the transformer model and derive the energy function whose gradients yield a layer which is very close in structure to a transformer layer. Let $\mathbf{x} \in \mathbb{R}^{D \times N}$ be an input sequence of length $N$ embedded in $D$ dimensions. We will denote its components by $\mathbf{x}_{Ai}$ with $A = 1\dots N$ and $i = 1\dots D$, or $\mathbf{x}_A$ suppressing the embedding index, but keeping the token index. Let $\mathbf{x}^{(t)}$ be the output sequence of layer $t$ of the model with $\mathbf{x}^{(0)} = \mathbf{x}$. A conventional transformer layer has an Attention layer (AT) followed by a two-layer feedforward (FF) and LayerNorm (LN) in series

$$\mathbf{x}^{(t+1)} = \mathbf{x}^{(t)} + \text{FF}\Big[\text{LN}\Big(\mathbf{x}^{(t)} + \text{AT}\big(\text{LN}(\mathbf{x}^{(t)})\big)\Big)\Big] \tag{4}$$

But subsequent works such as GPT-J (Wang and Komatsuzaki, 2021), PaLM (Chowdhery et al., 2023), and Energy Transformer (Hoover et al., 2023) showed that the following parallel design has good performance too

$$\text{Parallel Transformer:} \quad \mathbf{x}^{(t+1)} = \mathbf{x}^{(t)} + \text{AT}(\mathbf{g}^{(t)}) + \text{FF}(\mathbf{g}^{(t)}), \quad \mathbf{g}^{(t)} = \text{LN}(\mathbf{x}^{(t)}) \tag{5}$$

We choose this parallel transformer design as it is more suitable for our goal of replacing the transformer layer with the gradient of an energy.

If passing through a layer becomes one step of energy decent (ED), then all layers need to share weights. Therefore, our model will consist of a single module replacing the transformer block. Instead of different layers, we will be recurrently feeding the output of the layer back into itself, so that $\mathbf{x}^{(t)}$ will become step $t$ of the ED instead of the layer number.

**Update Rule** An important point to note is that the update rule of NRGPT is slightly different from conventional gradient descent and is of the form

$$\dot{\mathbf{x}} = \mathbf{x}^{(t+1)} - \mathbf{x}^{(t)} = -\boldsymbol{\eta}^{(t)} \frac{\partial E}{\partial \mathbf{g}^{(t)}} \tag{6}$$

where $\boldsymbol{\eta}^{(t)} \in \mathbb{R}^{D \times D}$ is an inference rate matrix, which can be learnable. Nevertheless, this can be a valid descent on $E$ as we can show that $E^{(t+1)} - E^{(t)} < 0$ under certain conditions, which depend on the normalization operation $\mathbf{g}$. We will derive these conditions for LayerNorm as well as RMSNorm, as well as when $\mathbf{g} = \mathbf{x}$, i.e., no normalization in Section 2.2. Next, we introduce the energy of NRGPT module.

## 2.1 ENERGY OF NRGPT

Matching our update rule (6) to the parallel transformer (5), we define two terms in the energy, $E^{\text{AT}}$ and $E^{\text{FF}}$

$$E = E^{\text{AT}} + E^{\text{FF}}, \qquad \boldsymbol{\eta}\partial_g E^{\text{AT}} = -\text{AT}(\boldsymbol{g}), \qquad \boldsymbol{\eta}\partial_g E^{\text{FF}} = -\text{FF}(\boldsymbol{g}), \tag{7}$$

We begin by introducing the attention layer and deriving the energy function for the self-attention mechanism. Then, we derive the energy function for the FF. Finally, we combine the two energy functions to obtain the total energy function for the transformer layer.

**Attention.** Consider a multi-head attention module with $H$ heads, and hidden dimension $Y = D/H$, index $h$ enumerates heads and runs $h = 1 \ldots H$. Its query, key, value and projection weights are

$$\boldsymbol{W}^Q, \boldsymbol{W}^K, \boldsymbol{W}^V, \boldsymbol{W}^P \in \mathbb{R}^{H \times Y \times D}, \tag{8}$$

Using the standard $\boldsymbol{K} = \boldsymbol{W}^K \mathbf{g}, \boldsymbol{Q} = \boldsymbol{W}^Q \mathbf{g}, \boldsymbol{V} = \boldsymbol{W}^V \mathbf{g}$, The MHA output for token $A$ is[1]

$$\text{AT}(\mathbf{g})_A = \sum_{h=1}^{H} \left[ \boldsymbol{W}_h^P \right]^T \boldsymbol{V}_h \text{SM} \left( \boldsymbol{K}_h^T \boldsymbol{Q}_{Ah} \right) \tag{9}$$

denoting $\boldsymbol{J} = \left[ \boldsymbol{W}^K \right]^T \boldsymbol{W}^Q$, the softmax is defined as (we omit the self-interaction term $C = A$)

$$\text{SM}(\boldsymbol{K}^T\boldsymbol{Q})_{BA} = \frac{\exp\left(\beta \boldsymbol{g}_B^T \boldsymbol{J} \boldsymbol{g}_A\right)}{\sum_{C<A} \exp\left(\beta \boldsymbol{g}_C^T \boldsymbol{J} \boldsymbol{g}_A\right)}, \quad \beta = \frac{1}{\sqrt{Y}} \tag{10}$$

Following Hoover et al. (2023), define the attention energy

$$E_A^{\text{AT}}(\boldsymbol{g}) = -\frac{1}{\beta} \sum_h \alpha_h \log \left[ \sum_{B<A} \exp\left(\beta \boldsymbol{g}_B^T \boldsymbol{J}_h \boldsymbol{g}_A\right) \right] \tag{11}$$

where $\boldsymbol{\alpha} \in \mathbb{R}^H$ is a learnable weight.

---

[1]Usually the projection weights $\boldsymbol{W}^P$ are defined as $D \times D$ and the head outputs are concatenated, into an $N \times (YH) = N \times D$ matrix before multiplying by $W^P$. This is equivalent to our definition.

Taking the gradient of $E^{\text{AT}}$ w.r.t. $\boldsymbol{g}_A$ and using (7), the resulting attention layer becomes

$$\text{AT}(\boldsymbol{g})_A = -\boldsymbol{\eta} \frac{\partial E_A^{\text{AT}}(\boldsymbol{g})}{\partial \boldsymbol{g}_A} = \sum_{h=1}^{H} \alpha_h \boldsymbol{\eta} \boldsymbol{J}_h^T \boldsymbol{g} \text{SM}\left(\boldsymbol{g}^T \boldsymbol{J}_h \boldsymbol{g}_A\right). \tag{12}$$

Both standard attention and this update have the form $\text{AT}(\mathbf{g}) = \boldsymbol{M}\mathbf{g}\text{SM}(\mathbf{g}^T \boldsymbol{J}\mathbf{g})$, but in the standard attention $\boldsymbol{M} = [\boldsymbol{W}^P]^T \boldsymbol{W}^V$ and in NRGPT attention is $\boldsymbol{M} = \alpha\boldsymbol{\eta}\boldsymbol{J}^T$. That is,

$$\text{Original: } \left[\boldsymbol{W}_h^P\right]^T \boldsymbol{W}_h^V \equiv \text{Energy: } \alpha_h \boldsymbol{\eta} \boldsymbol{J}_h^T \tag{13}$$

In principle $\boldsymbol{W}^V$ and $\boldsymbol{W}^P$ can be merged into one matrix. (He and Hofmann, 2024) also experimented with removing $\boldsymbol{W}^V$ and $\boldsymbol{W}^P$ and found that in the setting without skip connections, these two weights could be largely omitted.

**Feed-Forward network.** The FF network generally has two layers $\text{FF}(\boldsymbol{g}_A) = \boldsymbol{W}^{2T}\sigma\left(\boldsymbol{W}^1\boldsymbol{g}_A\right)$ with weights $\boldsymbol{W}^1, \boldsymbol{W}^2 \in \mathbb{R}^{M \times D}$, with $M$ being the size of the hidden layer. A possible choice for this network is a Dense Associative Memory (Krotov and Hopfield, 2016). In this case

$$E^{\text{FF}} = -\sum_{A=1}^{N} \mathbf{1}^T F\left(\boldsymbol{W}^1 \boldsymbol{g}_A\right), \quad \text{s.t. } F' = \sigma$$

$$\text{FF}(\boldsymbol{g}_A) = -\eta \frac{\partial E^{\text{FF}}}{\partial \boldsymbol{g}_A} = \boldsymbol{\eta}\boldsymbol{W}^{1T}\sigma\left(\boldsymbol{W}^1 \boldsymbol{g}_A\right) \tag{14}$$

where $\mathbf{1}$ is an $M$-dimensional vector of ones. Both the standard MLP in transformers and the FF update in Equation (14) have the form $\boldsymbol{M}\sigma(\boldsymbol{W}^1 g)$. In the standard MLP, $\boldsymbol{M} = \boldsymbol{W}^2$, whereas in NRGPT we get $\boldsymbol{M} = \boldsymbol{W}^1\boldsymbol{\eta}^\top$. That is,

$$\text{Original: } \boldsymbol{W}^2 \equiv \text{Energy: } \boldsymbol{W}^1\boldsymbol{\eta}^T. \tag{15}$$

As an example, in order for $E^{\text{FF}}$ to reproduce the FF of transformers with $\sigma(z) = \text{ReLU}(z) = \max(z, 0)$, the function $F$ should be

$$F(z) = \frac{1}{2}\sigma(z)^2 \tag{16}$$

Of course, the FF module can be replaced by other, more general, MLP networks. Essentially, any scalar function, which is additive in token index, can serve as a valid form of FF network. In the experiments (Section 3) we will detail our choices of $E^{\text{FF}}$.

**Distinction from prior work** NRGPT's energy is distinct from prior works like that of Energy Based Transformers (EBT) (Gladstone et al., 2025) and Energy Transformer (ET) (Hoover et al., 2023). Specifically, EBT computes energies of next tokens an output of a standard transformer forward pass, while NRGPT describes a parameterized energy whose *gradient* returns a transformer block. Repeatedly minimizing the energy by following this gradient resembles a complete transformer forward pass. On the other hand, ET's architecture appears more similar to NRGPT's design, but it's design was unsuited for *causal* token generation. We expand on these differences further in Appendix B.

## 2.2 Normalization of tokens

These normalizations have the form

$$\boldsymbol{g} = \boldsymbol{\gamma} \odot \frac{\boldsymbol{x} - \boldsymbol{\mu}}{\sqrt{\frac{1}{D}\|\boldsymbol{x} - \boldsymbol{\mu}\|^2 + \epsilon}} + \boldsymbol{\delta} \tag{17}$$

with $\boldsymbol{\mu} = \mathbb{E}[\boldsymbol{x}]$ for LayerNorm, and $\boldsymbol{\mu} = 0, \boldsymbol{\delta} = 0$ for RMSNorm. Here $\boldsymbol{\gamma}, \boldsymbol{\delta} \in \mathbb{R}^D$ and $\odot$ is elementwise multiplication. Many recent models such as Qwen Bai et al. (2023) and Llama Grattafiori et al. (2024) use RMSNorm Zhang and Sennrich (2019).

**Proposition 2.1** (Energy Descent). *The update rule (6) results in asymptotically decreasing energy,* $\dot{E}_A = E_A^{(t+1)} - E_A^{(t)} < 0$, *if the inference rate is* $\boldsymbol{\eta} = c \operatorname{diag}(\gamma)$ *with* $c \in \mathbb{R}_{>0}$. *This asymptotic behavior begins after a transient phase where previous tokens are converging.*

*Sketch of proof.* See Appendix E for full proof. The Jacobian of $\boldsymbol{g}_A$ can be written as $\partial \boldsymbol{g}_A / \partial \boldsymbol{x}_A = \frac{1}{r_A} \boldsymbol{\Gamma} \boldsymbol{P}_A$, where $\boldsymbol{\Gamma} = \operatorname{diag}(\gamma)$, $r_A > 0$ is some norm of $\boldsymbol{x}_A$ and $\boldsymbol{P}_A$ is an approximate $D \times D$ projection matrix and p.s.d. Using this and (6), once $\dot{\boldsymbol{x}}_B = 0$ for $B < A$, we get $\dot{E}_A = -r_A^{-1} \sum_A \operatorname{Tr} \left[ \partial_{\boldsymbol{g}_A} E_A \boldsymbol{\Gamma} \boldsymbol{P}_A \boldsymbol{\eta}^T \partial_{\boldsymbol{g}_A} E^T \right]$. When $\boldsymbol{\eta} = \boldsymbol{\Gamma}$ we get $\dot{E} < 0$. $\square$

There also exist $x$-dependent solutions of the form $\boldsymbol{\eta} = \boldsymbol{M}(\boldsymbol{x}) \boldsymbol{P}_A \boldsymbol{\Gamma}$ where $\boldsymbol{M}(\boldsymbol{x})$ is an arbitrary positive semi-definite (psd) matrix and $c > 0$, but they mean $\boldsymbol{\eta}$ is itself a neural network. To remain close to the structure of conventional transformers, we work with the $x$-independent $\boldsymbol{\eta} = c\boldsymbol{\Gamma}$ solution. In principle, $\boldsymbol{\eta}$ can also have a part contributing to the anti-symmetric part of $\boldsymbol{\Gamma} \boldsymbol{P}_A \boldsymbol{\eta}^T$, but we could not find an $x$-independent solution for it. While $\boldsymbol{\eta}^{(t)} = c^{(t)} \boldsymbol{\Gamma}$ puts severe restrictions on the preconditioning matrix, each layer is still allowed to have a different $c^{(t)}$ constant. Yet, as we show in Appendix E, LayerNorm together with Lipschitz activation functions in FF (e.g. ReLU) will lead to a bounded energy, which cannot explode. In this case, a generic $\boldsymbol{\eta}$ may still yield convergence. In fact, only a small set of $\boldsymbol{\eta}$ which yield perpetually oscillating solutions may not lead to convergence, although the exact conditions need to be derived.

**Preconditioner without layer normalization.** Several works have shown careful initialization and scaling can replace normalization entirely (Huang et al., 2020; Wang et al., 2024). Doing so would remove much of the restrictions on the preconditioner $\eta$. In an $T$-layer transformer, Instead of layer normalization, T-Fixup (Huang et al., 2020) and DeepNet (Wang et al., 2024) initialize some weights and $x$ by a scale proportional to $T^{-1/4}$ and change most layer outputs to $f(g) \to \alpha f(x)$, with $\alpha \propto T^{-1/2}$. In this case, $\eta$ becomes much less restricted.

**Proposition 2.2** ($\dot{E}$ without normalization). *When* $\boldsymbol{g} = \boldsymbol{x}$, *to get* $\dot{E} < 0$, *the symmetric part,* $\boldsymbol{\eta}_+ = (\boldsymbol{\eta} + \boldsymbol{\eta}^T)/2$ *needs to be psd.*

*Proof.* In this case $\dot{\boldsymbol{x}} = -\boldsymbol{\eta} \partial_{\boldsymbol{x}} E$ and

$$\dot{E} = -\sum_A \operatorname{Tr} \left[ \partial_{\boldsymbol{x}_A} E \boldsymbol{\eta} \partial_{\boldsymbol{x}_A} E^T \right] = -\sum_A \operatorname{Tr} \left[ \partial_{\boldsymbol{x}_A} E \boldsymbol{\eta}_+ \partial_{\boldsymbol{x}_A} E^T \right] \tag{18}$$

which is negative when $\boldsymbol{\eta}_+$ is psd. $\square$

The antisymmetric part $\boldsymbol{\eta}_- = (\boldsymbol{\eta} - \boldsymbol{\eta}^T)/2$ is not constrained by this and can be arbitrary at this point. Hence, without layer normalization, we can have $\boldsymbol{\eta}^{(t)}$ as learnable parameters of the form

$$\boldsymbol{\eta} = \boldsymbol{U}^T \boldsymbol{U} + \boldsymbol{V} - \boldsymbol{V}^T, \qquad\qquad \boldsymbol{U} \in \mathbb{R}^{D \times D'}, \boldsymbol{V} \in \mathbb{R}^{D \times D} \tag{19}$$

where $D'$ can be arbitrary, and each layer (depth) can have a different learnable $\boldsymbol{U}^{(t)}, \boldsymbol{V}^{(t)}$.

## 2.3 Asymptotic Stability of NRGPT

A peculiar aspect of NRGPT is the phenomenon of asymptotic stability. In order to illustrate it, consider a simplifying case when the inference rate matrix $\boldsymbol{\eta}$ is identity. In this case dynamical equations for tokens can be written as

$$\dot{x}_{iA} = -\frac{\partial E_A}{\partial g_{iA}} \tag{20}$$

Additionally, $\gamma$ can always be absorbed into $\boldsymbol{J}$ and the weights of the FF model. This way, the Jacobian becomes $\partial \boldsymbol{g}_A / \partial \boldsymbol{x}_A = \boldsymbol{P}_A$, which is p.s.d.. The key observation is that due to causal attention mask the energy $E_A$ of token $A$ only depends on the states of tokens $B \le A$. Thus, for the first token

$$\dot{\boldsymbol{x}}_1 = -\frac{\partial E_1}{\partial \boldsymbol{g}_1} \tag{21}$$

Since the energy of that token decreases with time

$$\dot{E}_1 = \frac{\partial E_1}{\partial \boldsymbol{g}_1} \frac{\partial \boldsymbol{g}_1}{\partial \boldsymbol{x}_1} \frac{d\boldsymbol{x}_1}{dt} = -\dot{x}_1^T \boldsymbol{P}_1 \dot{x}_1 \leq 0 \tag{22}$$

since $\boldsymbol{P}_1$ is psd. Additionally, since energy only depends on $\boldsymbol{g}$ – layernormalized tokens – it is bounded from below. Thus, the dynamics of $\boldsymbol{g}$ has to converge to a fixed point. This means that after a transitory period of time $T_{\text{tr}}$ the derivative $\frac{d\boldsymbol{g}_1}{dt}$ vanishes.

Now, consider the network of two tokens

$$\dot{E}_2 = \frac{\partial E_2}{\partial \boldsymbol{g}_1} \frac{\partial \boldsymbol{g}_1}{dt} + \frac{\partial E_2}{\partial \boldsymbol{g}_2} \frac{\partial \boldsymbol{g}_2}{\partial \boldsymbol{x}_2} \frac{d\boldsymbol{x}_2}{dt} = -\dot{x}_2^T \boldsymbol{P}_2 \dot{x}_2 \leq 0 \tag{23}$$

the second equality is written assuming that we are looking at this quantity at $t > T_{\text{tr}}$. Thus the first term is zero. Same argument applies, the energy decreases with time and is bounded from below. Thus, eventually $\boldsymbol{g}_2$ freezes.

One can apply this argument recursively to each token and conclude that after a transitory period of time, all tokens stabilize and all $\boldsymbol{g}_A$ will eventually become constants. From the perspective of energy profiles, this leads to the following behavior: during transitory regime energies of individual tokens will evolve in time (they can both increase and decrease). After that transitory period is over the energies must stabilize and reach their constant values that become unchanged in the future. One can run the inference dynamics as long as desired after that, but no changes in energies will occur. This behavior is apparent from the numerical profiles of energies, see Figure 2. It is a distinct aspect of our models - the phenomenon that we call asymptotic stability.

## 3 EXPERIMENTS

We tested our model on three datasets: ListOps, Shakespeare and Open Web Text (OWT). The details of the experimental settings and hyperparameters can be found in Appendix F. To evaluate the quality of text in the Shakespeare and OWT experiments, we use a number of metrics, including perplexity and diversity scores, explained in Appendix F.2.

**Model Variants.** The choice of the energy function is equivalent to the choice of architectures in neural networks. As our goal is to be as close to the original transformer architecture as possible, we experimented with a few settings for the FF network and found the following variants to be the best performing:

1. `NRGPT_H_FF1`: $E^{\text{FF}} = -\|\sigma(\boldsymbol{W}\boldsymbol{g}_A)\|^2$, same as in equation 14, but with $\sigma = \text{GELU}$.

2. `NRGPT_H_FF2W`: $E^{\text{FF}} = -\sum_A \boldsymbol{g}_A^T \boldsymbol{W}^2 \sigma(\boldsymbol{W}^1 \boldsymbol{g}_A)$, which yields

$$\text{FF}(\boldsymbol{g}_A) = -\eta \left( \boldsymbol{W}^2 \sigma(\boldsymbol{W}^1 \boldsymbol{g}_A) + \sigma'(\boldsymbol{W}^1 \boldsymbol{g}_A)^T \odot \boldsymbol{W}^{1^T} \sigma(\boldsymbol{W}^2 \boldsymbol{g}_A) \right) \tag{24}$$

Here, the first term is the conventional FF of transformers, but the second is a somewhat odd network.

In case with two weights, we choose the hidden dimension between $\boldsymbol{W}^1$ and $\boldsymbol{W}^2$ to be $4 \times D$. All of these performed well, with the residual version showing best performance on ListOps, learning at even smaller sizes than our baseline, while also easily training at larger sizes with embedding size over 256. However, since some of these models deviate significantly from the FF of a recurrent GPT (the gradient results in an FF module with four layers), we decided to focus more on `NRGPT_H_FF1` and `NRGPT_H_FF2W`, which are much closer to the GPT FF. As Baselines, we used `GPT_Rec_parallel`, which is a GPT-J model with a single transformer layer, feeding back recurrently into itself for a fixed number of times (mimicking number of layers). On Shakespeare and OWT, we also show results of a conventional GPT2-style deep transformer model.

### 3.1 ENERGY DYNAMICS

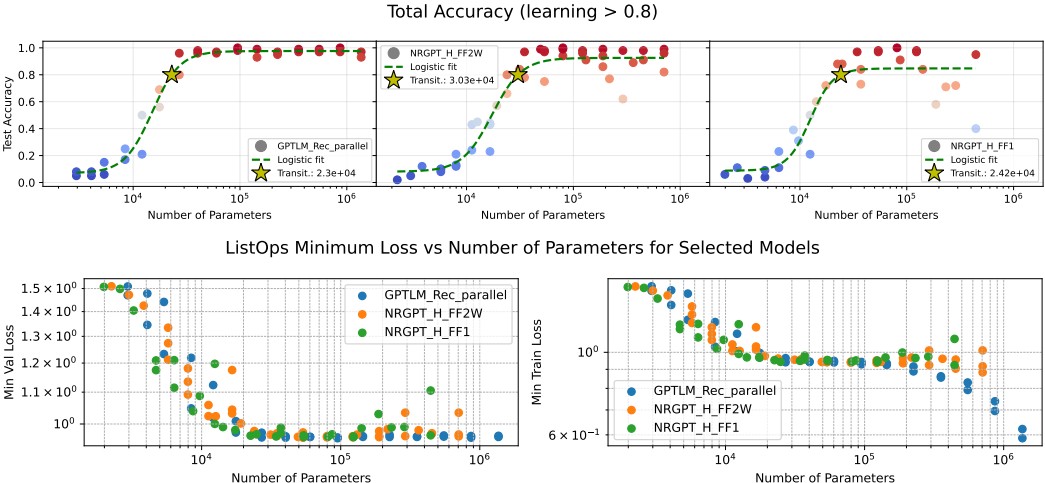

Figure 3: **Learning ListOps:** NRGPT variants match performance with a recurrent GPT model on ListOps accuracy parameter-transition points (top) and training/validation losses (bottom). The accuracy of models is tested on nested, mixed arithmetic tasks of maximum, median and sum modulo 20. For all plots, the x axis shows the *total parameter count* of the model. The yellow star indicates the transition to learning, which we define as where the logistic fit hit $> 80\%$ accuracy. The baseline model `GPT_Rec_parallel` shows the earliest learning transition at size $2.3 \times 10^4$, but our NRGPT variants are also similar, with `NRGPT_H_FF1` at $2.4 \times 10^4$ and `NRGPT_H_FF2W` at $2.98 \times 10^4$.

NRGPT without constraints on the inference rate $\eta$ is not forced to strictly decrease energy during inference and it may learn other exploration strategies for inference. Nevertheless, we would like to examine whether models which are explicitly forced to perform GD and reduce energy during inference can learn the tasks well. To better understand how our gradient-based update rule performs inference, we ran experiments on ListOps with large number of recurrent steps (30 steps). For these experiments, we set $\eta = 1$, which according to Proposition 2.1 forces the update rule to decreases energy. Figure 2 shows the evolution of total $E$, $E^{\mathrm{AT}}$ and $E^{\mathrm{FF}}$ along these trajectories. We observe that indeed in all trajectories the final energy is lower than initial. Each individual token trajectory is not required to be monotonically decreasing, as the dynamics of tokens is coupled. however, in accordance with our result in Section 2.3, after a transient stage, once all previous tokens start converging, the energy of the next token monotonically decreases.

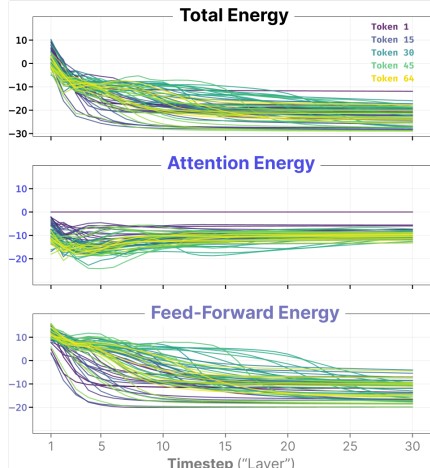

Figure 2: In NRGPT, tokens converge to stable states of low energy where the causal attention mask allows each token energy to fluctuate during inference. Shown are 64 tokens passed to an NRGPT model trained to predict ListOps equations.

## 3.2 LISTOPS

We perform experiment on nested math operations on lists of integers, which are a version of ListOps (Nangia and Bowman, 2018). Our ListOps setting consists of three functions: maximum, median and sum modulo 20. Our inputs range from 0 to 19. Each data sample begins with nested equations like `SUM(2,MAX(4,13,1),MEDIAN(5,3,16))`. As performance metrics, we looked at accuracy on the mixed task, as well as the training and validation loss. Figure 3 shows the results for two of our model variants, `NRGPT_H_FF1` and `NRGPT_H_FF2W` as compared to `GPT_Rec_parallel`.

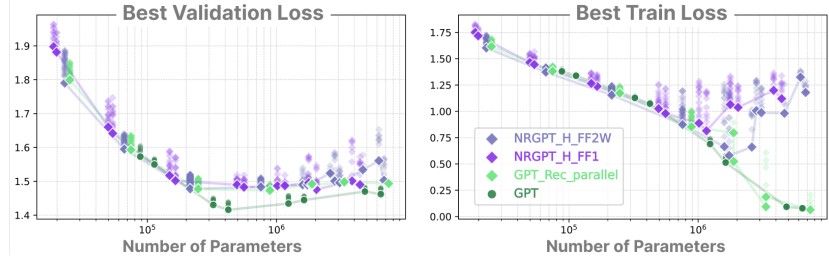

Figure 4: **Shakespeare scaling:** NRGPT achieves performance parity with recurrent GPT on Shakespeare across parameter sizes, as measured by *best validation loss* per number of parameters. For many embedding sizes, NRGPT also follows the same optimal training loss trajectory-per-parameter as both GPT and recurrent GPT baselines. However, NRGPT does not overfit Shakespeare at large parameter sizes. Connecting lines show the best performance at fixed parameter sizes. Transparent dots show different choices of hyperparameters — a larger spread indicates more sensitivity to hyperparameters. See Appendix F.4 for details.

Table 1: OWT performance at $n_{embed} = 768$.

| **Training loss** | | | | | |
|---|---|---|---|---|---|
| **Model** | mean $\pm$ std | min | max | # runs | # Param. |
| GPT | **2.905 $\pm$ 0.006** | **2.900** | **2.914** | 5 | 124M |
| GPT_Rec_parallel | 3.447 $\pm$ 0.046 | 3.395 | 3.494 | 5 | 85M |
| NRGPT_H_FF2W | 3.456 $\pm$ 0.076 | 3.391 | 3.540 | 3 | 90M |

| **Validation loss** | | | | | |
|---|---|---|---|---|---|
| **Model** | mean $\pm$ std | min | max | # runs | # Param. |
| GPT | **2.921 $\pm$ 0.005** | **2.915** | **2.929** | 5 | 124M |
| GPT_Rec_parallel | 3.454 $\pm$ 0.037 | 3.411 | 3.491 | 5 | 85M |
| NRGPT_H_FF2W | 3.467 $\pm$ 0.073 | 3.404 | 3.548 | 3 | 90M |

## 3.3 SHAKESPEARE

We compare the performance of our `NRGPT_H_FF2W` and `NRGPT_H_FF1` with `GPT_Rec_parallel` and deep GPT for embeddings sizes less than 1024 (Figure 4 In larger sizes, we ran many sweeps to find suitable hyperparameters such as the range of learning rates, resulting in the wide spread. Interestingly, our best models at large sizes achieve validation losses similar to the GPT baselines, but do so with much less overfitting to the training set.

## 3.4 OPEN WEB TEXT

Table 2 shows the best model configuration for baseline GPT, RGPT-parallel, and our model NRGPT with the respective generation quality metrics. We see that the generation quality of NRGPT is very competitive with GPT and RGPT-parallel while it contains around 34M less parameters than GPT. Figure 5 shows example of generated text by GPT, RGPT-parallel and NRGPT for which the generation quality metrics are provide in Table 2. In these experiments, we consider transformer blocks configured at the same *width* — please see Appendix F.5 for more experiments across equal parameter counts, including experiments that show NRGPT's advantage on downstream tasks such as MMLU (Hendrycks et al., 2020).

## 4 LIMITATIONS

NRGPT is an appealing theoretical construct for the inference process of GPT. In our experiments, we observe that NRGPT can achieve similar performance to GPT and its recurrent variants on ListOps,

Table 2: Best Model Configurations and Quality Metrics for OWT. Note abbreviations: no of parameters → n_param, grammar quality score → gqs, average pariwise cosine similarity → apcs, distinct-1 → d-1 and distinct-2 → d-2.

| Model | Configuration | | | | | | Metrics | | | | |
|---|---|---|---|---|---|---|---|---|---|---|---|
| | lr | min_lr | n_layer | n_head | n_embed | n_params | perplexity | gqs | apcs | d-1 | d-2 |
| GPT | 7e-4 | 7e-5 | 12 | 12 | 768 | 124M | **75** | **0.978** | 0.306 | 0.619 | 0.965 |
| GPT_Rec_Parallel | 6e-4 | 4e-4 | 12 | 12 | 768 | **85M** | 99 | 0.976 | **0.336** | 0.615 | 0.975 |
| NRGPT_H_FF2W | 1e-4 | 7e-5 | 12 | 12 | 768 | 90M | 104 | 0.966 | 0.306 | **0.674** | **0.984** |

Shakespeare, and OWT. However, NRGPT is computationally the gradient of an energy, which enforces weight sharing and limits how flexibly we can parameterize the architecture. We observe that this constraint also causes a larger amount of hyperparameter sensitivity than GPT variants. We also notice that the configurations of NRGPT that achieve competitive performance at comparable parameters actually have a slightly larger FLOP count than their standard transformer counterparts. Please see our extended discussion in Appendix D. In contrast to standard transformers, increasing the number of attention heads in NRGPT actually increases the parameters. We additionally observe that NRGPT has a more difficult time overfitting the training set, which is beneficial in small data regimes but is undesirable in the massive datasets used to train modern LLMs.

## 5 DISCUSSION AND CONCLUSIONS

We have presented NRGPT, a minimal modification of the GPT architecture that unifies autoregressive language modeling with energy-based modeling. Our analysis shows that under specific conditions on the inference rate matrix $\eta$, this process provably decreases energy after a transient period of time, providing a principled foundation for the dynamics, a phenomenon that we call asymptotic stability. Moreover, relaxing this constraint allows the model to learn its own energy exploration strategy for inference. Thus, our work complements previous studies suggesting that transformers perform GD during inference. Unlike past work, in our model inference is explicitly a gradient-based dynamics, while still maintaining an architecture very similar to GPT. Our experiments show that this framework performs comparably to a fully recurrent GPT model across parameter sizes while generally requiring fewer parameters. NRGPT represents a meaningful step toward understanding the architecture of transformers using energies.

## 6 REPRODUCIBILITY STATEMENT

The code for our model, experiments, and analysis is available at https://github.com/bhoov/nrgpt in a self-contained environment. The code began as a fork of the excellent nanoGPT repository and as such all experiments and models are implemented using PyTorch (Paszke et al., 2019). Each Shakespeare and ListOps experiment was conducted on a single H100 GPU. OWT experiments were conducted over 4 nodes of 8xH100 GPUs.

## ACKNOWLEDGMENTS

The results presented here were obtained while Dmitry Krotov was employed by IBM Research. At the time of the camera-ready submission Dmitry Krotov is no longer employed by IBM Research.

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

# A    LLM USAGE STATEMENT

LLMs were not used for ideation, experiments, or model design. LLM generated code helped with experimental analysis (e.g., plot layouts) and grammar checking of the submission.

# B    NRGPT COMPARED TO EBT AND ET

## B.1    DISTINCTION FROM EBT

Both the Energy-Based Transformers (EBT) of (Gladstone et al., 2025) and NRGPT are methods that can be used to minimize an explicit energy during autoregressive generation. However, the methods differ in *how* that energy is modeled. Specifically, the EBT paper uses standard, feed-forward transformer architectures with modified attention masks and a standard loss function on predicted tokens to turn the transformer predictions into an energy, whereas we use a novel, causal energy formulation of the transformer block inspired by the Energy Transformer (Hoover et al., 2023).

NRGPT is most comparable to the autoregressive EBT. The EBT paper mentions that "the autoregressive EBT presents greater implementation challenges, primarily due to the potential for information leakage in naïve implementations", and they discuss the design challenges in Sec C.3 of their Appendix, which requires reframing how tokens are processed by the standard transformer architecture. To prevent information leakage, they duplicate a sequence of tokens into *observed* tokens and *predicted* tokens while carefully tuning both the attention masks and the contraction over observed values.

We believe that NRGPT's solution for a causal EBM is more succinct and elegant than EBT's, preventing duplicate tokens and the need for careful tuning of attention masks. Our solution is expressed in Equation (3), where $E_A$ describes the energy to predict token $A$. By restricting the summation of the `logsumexp` to include summation only over previous tokens $B < A$, and then by minimizing the energy w.r.t. token $\mathbf{g}_A$, we are guaranteed to propagate information causally without any of the engineering tricks of EBT.

## B.2    DISTINCTION FROM ET

The architecture of Energy Transformer (ET) (Hoover et al., 2023) is more similar to that of NRGPT, but it is fundamentally different. This is because ET is designed for MASKed-token prediction tasks whereas NRGPT is a special energy function designed for *autoregressive token prediction*, a paradigm that is not compatible with the original ET's bidirectional attention design (where attention signal propagates both forward and backwards in time such that past tokens can attend to future tokens). A simple causal mask on ET attention breaks ET's guarantee of monotonic energy minimization.

The key innovation of NRGPT is to model the sequence as a collection of token-wise energies $E_A$ for token index $A$, rather than a global sequence energy $E = \sum_A E_A$. We discover that tokens still converge even when all tokens minimize their individual energy simultaneously (see Figure 2), and we argue that this generalizes the ideas of ET to allow tokens to explore a meaningful energy landscape during causal token generation.

We also theoretically and empirically study the **projection matrix** that is present in all standard transformer attention but that is *noticeably absent* in ET's attention formulation. We interpret this matrix as an "inference rate" matrix $\boldsymbol{\eta}$ in the gradient descent step, where under certain conditions we are guaranteed to maintain token convergence. Including this projection matrix and playing with various choices for $E^{\mathrm{FF}}$ of Equation (3) makes the NRGPT block *as expressive* as a recurrent, standard GPT block that can arguably go toe-to-toe with similarly configured GPT models on causal language modeling tasks up to the size of OWT (see Table 5 and Table 4). These conclusions and experiments were not evident in the original ET paper.

In summary, NRGPT is distinct from ET because:

1. **NRGPT performs causal language modeling by minimizing a per-token energy**. ET was restricted to strict energy minimization of an entire sequence, a paradigm that is not compatible with the parallel, autoregressive language modeling of GPT-style transformers.

2. **NRGPT uses learnable inference rate matrices $\eta$** during token prediction. Meanwhile, ET was restricted to a fixed, scalar gradient descent step which did not allow additional exploration of the energy landscape.

3. **NRGPT explores alternative energy-replacements for the feed-forward (FF) MLP module**. ET used a single-layer Hopfield Network with energy $G(\boldsymbol{\xi}\mathbf{g}_A)$, which results in the weights of the two layers to be $\boldsymbol{\xi}$ and $\boldsymbol{\xi}^T$. In NRGPT, we explore a more general form $E^{\mathrm{FF}}$ (e.g., Equation (24)) for the feed-forward module and find improved results on the causal language modeling task.

## C   ADVANTAGES OF EBMs

EBMs offer a paradigm for generative modeling that is inherently about optimization, where the model samples a new token from a **transition probability** described by the prior context. Conventional transformers are trained to *implicitly* learn this transition probability, but it is hidden in the architectural design. The key appeal of an EBM (like NRGPT) is that it models the transition probability *explicitly*. This offers several advantages which we hope to explore in future work:

1. **Systematic exploration of the solution space**. An explicit likelihood function enables us to systematically explore the space of solutions in LLMs using well-established methods in optimization and statistical physics, such as alternative gradient descent methods, minimum-energy paths, saddle-point analysis, and metastability. These tools are not available when the energy is implicitly encoded in a deep architecture, and we believe that finding alternative or creative solutions may correspond to exploring different local minima of the energy during inference. It also means that we can now view the forward process of causal LMs like GPT as a formal optimization process, which despite the prevalence of research around *in-context learning* (ICL) (Dong et al., 2024), is not a widespread belief.

2. **Variable computation using early stopping criteria**. Both the energy and the norm of the energy's gradient can be used as signals to stop model computation "early" for easier problems, or to continue thinking longer for harder ones. This is a primary motivation of other explicit EBM language models like EBT (Gladstone et al., 2025), and we discuss this advantage further below.

3. **Model alignment using energy regularizers**. Note that Equation (7) of our paper shows NRGPT's architecture as the sum of two energies: a token-mixing *attention energy* $E^{\mathrm{AT}}$ and a token-wise *feed-forward energy* $E^{\mathrm{FF}}$. Per the precedent of ET (Hoover et al., 2023), these are chosen to be faithful to the original transformer's design. However, any scalar objective can theoretically be added to NRGPT's block, and we imagine that adding regularizer terms to bias the energy landscape toward favorable solutions (or away from undesirable ones) is a unique benefit of EBMs that is more robust to prompt injection attacks than current LLMs.

**Early stopping**   Adaptive stopping criteria for "early stopping" is an incredible advantage for EBMs over traditional methods in the context of language modeling. If you can guarantee that all of NRGPT's tokens $B < A$ are fixed, then the energy of token $A$ is guaranteed to decrease, and early stopping based on energy values and gradient norms (as you suggested) would just work — we are iteratively improving a token's embedding until the convergence guarantees are met and there is no purpose to "thinking longer". This iterative improvement needs no fine-tuning as it is baked into the model design. However, the constraint of a fixed preceiding context has a strong *disadvantage*: doing this would discard the transformer's unique advantage of full parallelism across tokens. For the causal energy formulation with parallel token evolution studied in this work, we are actually not guaranteed to monotonically decrease the energy and energy deltas and gradient norms would be unreliable (see Figure 2 for token-wise energy trajectories).

## D   ON FLOP COMPLEXITY

NRGPT is parameter efficient compared to standard GPT and recurrent GPT, but how does it compare on the FLOP cost of the model? When we consider the FLOPs/block (per iteration), we discover that NRGPT is anywhere from $1-2\times$ the FLOP cost of an equivalent RecGPT at constant parameters,

where the factor of 2 difference appears depending on which FF variant we use (`NRGPT_H_FF1` or `NRGPT_H_FF2W`). We expand on the contributions of our architectural choices below:

- **NRGPT Attention**: For a single head, the attention block of NRGPT costs approximately the same FLOPS as recurrent and normal GPT. In fact, NRGPT uses slightly fewer FLOPs since the key weights $\boldsymbol{J}\mathbf{g}_B$ are used for both the `keys` and the `values` of the attention update (in contrast to the standard attention where distinct values $\boldsymbol{W}^V\mathbf{g}_B$ are computed for each key $B$). For multiple heads however, NRGPT costs more because our current solution uses $D \times D$ weights $J_h$ *per head*, while GPT attention uses increasingly narrow KQV matrices as you increase the number of heads (i.e., the dimension of $\boldsymbol{W}^V$ is of shape $D \times (D/H)$, which decreases as $H$ becomes larger). This is a convention that can be easily adapted to NRGPT but we did not explore in this work.

- **NRGPT FF**: This FLOP count for this FF module can vary, but in this paper we tested configurations that are always $\sim 2\times$ the FLOP count of a standard transformer's FF of the same width. We discuss the two FF modules studied in this work below:

  1. **FF1**, where $E^{FF} = \|\sigma(\boldsymbol{W}\boldsymbol{g})\|^2$. In ListOps, we found that FF1 competes with Rec-GPT only when the hidden dimension of $\boldsymbol{W}$ was $8D$ instead of the standard $4D$ used in GPT. This makes FF1 a constant-parameter operation, but because the wider weight matrix is used twice it actually doubles the FLOPS cost of NRGPT.

  2. **FF2W**, where $E^{FF} = \boldsymbol{g}^T\boldsymbol{W^2}\sigma(\boldsymbol{W^1}\boldsymbol{g})$. In this config, the update rule $-\boldsymbol{\eta}\nabla E$ leads to two terms, each being a two layer neural net. We keep the $4D$ expansion used by standard transformers and thus the same parameter count, but this leads to a $\sim 2\times$ FLOPS in this module compared to standard transformers.

Thus, though NRGPT is more parameter efficient in general, it seems that parameter savings need to be exchanged for FLOPs. Note that we use pytorch's functional autograd interface to compute gradients through this network which slows down the wall-clock time evaluation compared to manual implementation of the update rule.

## E  LEARNING RATE AND PRECONDITIONER OF THE FORWARD PASS

$$\textbf{RMSNorm:} \quad g_{Ai} = \gamma_i \frac{x_{Ai}}{\sqrt{\frac{1}{D}\sum_i x_{Ai}^2}} = \sqrt{D}\gamma_i \hat{x}_{Ai} \tag{25}$$

$$\textbf{LayerNorm:} \quad g_{Ai} = \gamma_i \frac{x_{Ai} - \mathbb{E}[x_A]}{\sqrt{\mathrm{Var}[\mathrm{x_A}] + \epsilon}} + \beta_i \tag{26}$$

where $x_{Ai}$ are the components with $A \in 1 \ldots N$ and $i \in 1 \ldots D$.

*Proof of Proposition 2.1: Energy Descent.* Using chain rule

$$\dot{E}_A = \sum_{B,C} \frac{\partial E_A}{\partial \boldsymbol{g}_B} \frac{\partial \boldsymbol{g}_B}{\partial \boldsymbol{x}_C} \dot{\boldsymbol{x}}_C$$

$$= -\sum_{B,C} \frac{\partial E_A}{\partial \boldsymbol{g}_B} \frac{\partial \boldsymbol{g}_B}{\partial \boldsymbol{x}_C} \boldsymbol{\eta} \frac{\partial E_C}{\partial \boldsymbol{g}_C} \tag{27}$$

Defining $\Gamma = \mathrm{diag}(\gamma)$, the Jacobian of $g$ becomes

$$\frac{\partial \boldsymbol{g}_{Ai}}{\partial \boldsymbol{x}_{B\rho}} = \Gamma_{ij} \frac{\delta_{AB}}{r_A} \left(\delta_{j\rho} - y_{Aj}y_{A\rho}\right), \tag{28}$$

where

$$\textbf{RMSNorm:} \quad r_A = \|\boldsymbol{x}_A\|/\sqrt{D}, \qquad y_A = \boldsymbol{x}_A/r_A \tag{29}$$

$$\textbf{LayerNorm:} \quad r_A = \sqrt{\mathrm{Var}[\boldsymbol{x}_A] + \epsilon}, \qquad y_A = (\boldsymbol{x}_A - \boldsymbol{\mu}_A)/r_A. \tag{30}$$

Since typically $\epsilon = 10^{-5}$ is a small constant, $\|y_A\| \approx 1$ for LayerNorm, and exactly 1 for RMSNorm. Therefore

$$\frac{\partial \boldsymbol{g}_A}{\partial \boldsymbol{x}_A} = \frac{1}{r_A}\Gamma P_A, \qquad\qquad P_A = P_A^T, \qquad\qquad P_A^2 = P_A + O(\epsilon) \qquad (31)$$

where the approximate projection matrix $P_A$ is positive semi-definite, with $\hat{y}_A/\|y_A\|$ being its sole null eigenvector. Plugging into equation 27

$$\dot{E}_A = -\sum_B \frac{1}{r_B}\mathrm{Tr}\left[\frac{\partial E_A}{\partial g_B}\Gamma P_B \boldsymbol{\eta}^T \frac{\partial E_B}{\partial g_B}^T\right] \qquad (32)$$

Note that when $B > A$, $\partial E_A/\partial \boldsymbol{g}_B = 0$. Hence, equation 32 can be separated into $A = B$ and $B < A$

$$\dot{E}_A = \sum_{B<A} \mathrm{Tr}\left[\frac{\partial E_A}{\partial g_B}\frac{\partial \boldsymbol{g}_B}{\partial \boldsymbol{x}_B}\dot{\boldsymbol{x}}_B\right] - \frac{1}{r_A}\mathrm{Tr}\left[\frac{\partial E_A}{\partial g_A}\Gamma P_A \boldsymbol{\eta}^T \frac{\partial E_A}{\partial g_A}^T\right] \qquad (33)$$

In order for token $A$ to converge as well, we only need to find conditions under which the second term in equation 33 converges. This is because the same conditions would then lead the second term in all $\dot{E}_B$ for $B < A$ to converge. Then, by induction all $E_B$ will eventually converge. Thus we want just the second term

$$\delta E_A = -\frac{1}{r_A}\mathrm{Tr}\left[\frac{\partial E_A}{\partial g_A}\Gamma P_A \boldsymbol{\eta}^T \frac{\partial E_A}{\partial g_A}^T\right] \qquad (34)$$

to satisfy $\delta E_A < 0$, which can be achieved if the symmetric part of $\Gamma P_A \boldsymbol{\eta}^T$ is p.s.d.. Two simple solutions to this are

$$\boldsymbol{\eta} = c\Gamma, \qquad\qquad \text{or} \qquad\qquad \boldsymbol{\eta} = M(x)P_A\Gamma \qquad (35)$$

where $M(x)$ is an arbitrary p.s.d. matrix and $c > 0$. If we want $\boldsymbol{\eta}$ to be simple weights instead of an $x$ dependent neural network, the solution is $\boldsymbol{\eta} = c\Gamma$. $\qquad\square$

Note that equation 34 does not restrict the anti-symmetric part of $\Gamma P_A \boldsymbol{\eta}$. Using $\boldsymbol{\eta} = c\Gamma + B$, the antisymmetric part satisfies $B^T P_A \Gamma = -\Gamma P_A B$. Since $P_A$ is rank $D - 1$ for each $A$, the anti-symmetric part doesn't seem to have an $x$-independent solution. The fact that $P_A$ is different for each token severely restricts the form of $\boldsymbol{\eta}$ to get convergence. However, The boundedness of the energies $E_A$ due to boundedness of $\boldsymbol{g}_A$ can tame the dynamics and lead to more choices for $\boldsymbol{\eta}$ yielding convergence. For example, a damped harmonic oscillator has a dynamics in which the state oscillates in damped fashion, but the energy is constantly decreasing.

### E.1 EVOLUTION OF LOSS

We can always absorb $\Gamma$ into the weights of AT and FF, so we will assume $\Gamma = I$ from here on. To find $\dot{E}_A$ we need to use chain rule with derivatives w.r.t. all tokens $B < A$.

$$\begin{aligned}\dot{E}_A &= \sum_B \partial_{g_B}E_A \partial_B g_B \dot{x}_B \\ &= -\sum_B \partial_{g_B}E_A P_B \boldsymbol{\eta} \partial_{g_B}E_B \\ &= \sum_{B<A} \partial_{g_B}E_A^T P_B \dot{x}_B - \partial_{g_A}E_A^T P_A \boldsymbol{\eta}\partial_{g_A}E_A\end{aligned} \qquad (36)$$

because of the coupling between different tokens, it is not clear whether $\dot{E}_A$ is negative or not. Yet, due to the causality of the $B < A$ interaction, token $B$ is not affected by any future token. It follows that if token $B$ dynamics converges, meaning $\dot{x}_B \to 0$ for all $B < A$, we have

$$\text{if } \dot{x}_B = 0, \quad \forall B < A: \quad \dot{E}_A = \left(\frac{\partial E_A}{\partial g_A}\right)^T P_A \boldsymbol{\eta}\frac{\partial E_A}{\partial g_A} = -\dot{x}_A^T \boldsymbol{\eta}^{-1}P_A \dot{x}_A \qquad (37)$$

where the last part assumes $\boldsymbol{\eta}$ is invertible. Hence, token $A$ will converge if $P_A\boldsymbol{\eta}$ is p.s.d. Since $P_A$ is a projection away from $x_A$ and $P_A\boldsymbol{\eta}$ should be p.s.d. for all $A$, it follows that on the data manifold $\boldsymbol{\eta}$ can only be a constant times identity. If the the data is on a low-rank subspace, $\boldsymbol{\eta}$ can be arbitrary on the orthogonal subspace. More formally, let $P_\parallel$ represent projection onto $S_\parallel = \mathrm{Span}\{x_A | \forall A\}$, meaning $x_A^T P_\parallel x_A = \|x_A\|^2$. Let $P_\perp$ be the projection to the subspace orthogonal to all data $x_A$, meaning $x_A^T P_\perp x_A = 0$, and so $\|P_\perp P_\parallel\| = 0$. The $\boldsymbol{\eta}_\parallel = P_\parallel \boldsymbol{\eta} P_\parallel = cP_\parallel$ for some $c > 0$, while $\boldsymbol{\eta}_\perp = P_\perp \boldsymbol{\eta} P_\perp$ is an arbitrary p.s.d. matrix with null space including $S_\parallel$.

By induction, we can show that the $\boldsymbol{\eta}$ above leads to convergence of all tokens. For the first token $A = 1$, there are no past tokens to convergence requires

$$\dot{E}_1 = -\left(\frac{\partial E_1}{\partial g_1}\right)^T P_1 \boldsymbol{\eta} \frac{\partial E_1}{\partial g_1} < 0 \tag{38}$$

For the second token, when the first token converges, $\dot{x}_1 = 0$, resulting in the same condition for $\dot{E}_2$. Assume $\boldsymbol{\eta} = I$ and introduce the shorthand $E_{A;B} = \partial E_A / \partial g_B$, and the Mahalonobis norm $\|z\|_M^2 = z^T M z$. For any $A$

$$E_{A;A} P_A \dot{x}_A = -\|E_{A;A}\|_{P_A}^2 = -\|\dot{x}_A\|_{P_A}^2$$
$$\dot{E}_A = E_{A;A} P_A \dot{x}_A + \sum_{B<A} E_{A;B} P_B \dot{x}_B$$
$$= -\|\dot{x}_A\|_{P_A}^2 + \sum_{B<A} E_{A;B} P_B \dot{x}_B \tag{39}$$
$$\leq \sum_{B<A} E_{A;B} P_B \dot{x}_B \tag{40}$$

Since $E_A$ are bounded from below (shown next), $\dot{E}_A < 0$ should eventually lead to convergence. For $E_1$, there are no past tokens and $\dot{E}_1 < 0$ always, resulting in convergence.

### E.2 BOUNDEDNESS OF THE ENERGY.

First, observe that $E_A = E_A^{AT} + E_A^{FF}$ is bounded from below, given some assumptions about FF. For AT, since $g_B = \delta x_A / \|\delta x_A\| + \beta = \hat{y} + \beta$, we have

$$\|g_A\|^2 = 1 + 2\beta \cdot \hat{y}_A \qquad\qquad \Rightarrow 1 - 2\|\beta\| \leq \|g_A\|^2 \leq 1 + 2\|\beta\|$$
$$g_A^T \boldsymbol{J} g_B = \hat{y}_A^T \boldsymbol{J} \hat{y}_B + \beta \cdot (\boldsymbol{J}\hat{y}_B + \boldsymbol{J}^T \hat{y}_A)$$
$$- (1 + 2\|\beta\|)\|\boldsymbol{J}\|_2 \leq g_A^T \boldsymbol{J} g_B \qquad\qquad \leq (1 + 2\|\beta\|)\|\boldsymbol{J}\|_2 \tag{41}$$

where $\|\boldsymbol{J}\|_2$ is the spectral norm of $\boldsymbol{J}$ equal to the largest singular value of $\boldsymbol{J}$. Therefore, using monotonicity of log and exp

$$E_A^{AT} = \log \sum_{B<A} \exp[g_A^T \boldsymbol{J} g_B] \leq \log(A \max_{B<A}\{\exp[g_A^T \boldsymbol{J} g_B]\})$$
$$\leq \log A + \max_{B<A}\{g_A^T \boldsymbol{J} g_B\} \leq \log A + (1 + 2\|\beta\|)\|\boldsymbol{J}\|_2 \tag{42}$$

and similarly for the lower bound resulting in

$$-(1 + 2\|\beta\|)\|\boldsymbol{J}\|_2 \leq |E_A^{AT} - \log A| \leq (1 + 2\|\beta\|)\|\boldsymbol{J}\|_2 \tag{43}$$

For FF, we need to assume a form for the energy. First, considering $E^{FF}(g_A) = -\|\sigma(W g_A)\|^2$ with activation $\sigma$ being Lipschitz, as in ReLU or GeLU. The Lipschitz condition means

$$\|\sigma(x) - \sigma(y)\| \leq L\|x - y\| \tag{44}$$

for some Lipschitz constant $L > 0$. setting $y = 0$ and $x = W g_A$ we get

$$\|\sigma(W g_A) - \sigma(0)\| \leq L\|W g_A\| \leq L\|W\|_2\|g_A\| \leq L\|W\|_2(1 + \|\beta\|) \tag{45}$$

using triangle inequality $\|a + b\| \le \|a\| + \|b\|$, with $a = \sigma(Wg_A) - \sigma(0)$ and $b = \sigma(0)$ we get

$$\|\sigma(Wg_A)\| \le \|\sigma(Wg_A) - \sigma(0)\| + \|\sigma(0)\| \le L\|Wg_A\| + \|\sigma(0)\|$$
$$\le L\|W\|_2(1 + \|\beta\|) + \|\sigma(0)\| \tag{46}$$

and similarly, using the other side of triangle inequality $\|a\| \ge \|a + b\| - \|b\|$ with $a = \sigma(Wg_A)$ and $b = -\sigma(Wg_A) + \sigma(0)$ we have

$$\|\sigma(Wg_A)\| \ge \|\sigma(0)\| - \|\sigma(Wg_A) - \sigma(0)\| \ge \|\sigma(0)\| - L\|W\|_2(1 + \|\beta\|) \tag{47}$$

therefore

$$-(\|\sigma(0)\| + L\|W\|_2(1 + \|\beta\|))^2 \le E^{FF}(g_A) \le -(\|\sigma(0)\| - L\|W\|_2(1 + \|\beta\|))^2 \tag{48}$$

For more general FF, as long as they are MLP with Lipschitz activations, the normalized nature of $g_A$ should guarantee boundedness of $E^{FF}$ and, consequently, $E$.

# F EXPERIMENTS

## F.1 ARCHITECTURE DETAILS

NRGPT serves as a drop-in replacement for a standard GPT block, where tokens and positions are embedded using learnable embedding matrices following the precedent of GPT-2.[2] Text is tokenized using the default OpenAI BPE tokenizer from GPT-2 (with just over 50k vocab tokens). For prediction, tokens are "unembedded" using a linear projection layer whose weights are shared by the input's embedding matrix.

We describe the predictions of NRGPT as a *sampling process* obtained by minimizing the energy of each token, which serves as a recurrent replacement of the *complete forward pass* through the transformer. This is easiest to explain for when $\eta = 1$ where the forward pass is explicit gradient descent on the energy $E_A$ of token index $A$ (these energy trajectories are shown in Figure 2). Gradient descent (GD) provides a fast, differentiable way to move initial points to locations which have lower energy (higher likelihood). Each new token starts from an initial position $\mathbf{x}_A(t = 0)$, and the forward pass does GD, evolving it to a point $x_A(t = t_f)$ which has lower energy. In this case, $E_A$ can be interpreted as the log-transition probability. In all GPT-style models and therefore also in NRGPT, we choose the initial point $\mathbf{x}_A(t = 0)$ to be $\mathbf{x}_{A-1}$, the embedding of the previous token, but tokens could be initialized to anything in theory.

In the case where $\boldsymbol{\eta} \neq \mathbf{1}$, the forward pass follows more complex dynamics than pure GD, but in general the model learns dynamics that convert $\mathbf{x}_{A-1}$ to $\mathbf{x}_A$. We refer to this process as "sampling" throughout the paper; however, we note that this is not a strict sampling process for any value of $\boldsymbol{\eta}$.

## F.2 EVALUATION METRICS

To assess the generation quality of our language models, we utilize several complementary metrics. We use Perplexity as a measure of model uncertainty, computed using a pretrained GPT-2 model to evaluate how well the generated text aligns with expected language patterns. Lower perplexity indicates more fluent and predictable text, with scores typically ranging from 10 (excellent) to 1000+ (poor quality). For lexical diversity, we utilize Distinct-1 and Distinct-2, which measure the ratio of unique unigrams and bigrams to total n-grams (or total words) in the generated text, respectively. These metrics range from 0 to 1, where higher values indicate greater vocabulary diversity and less repetitive generation. A Distinct-1 score near 0 suggests highly repetitive text, while scores above 0.8 indicate rich vocabulary usage. We utilize Average Pairwise Cosine Similarity using Sentence-BERT embeddings (Reimers and Gurevych, 2019) to measure semantic diversity within generated samples. This metric calculates the mean cosine similarity between all pairs of generated sentences, ranging from -1 to 1. Optimal values fall between 0.3 and 0.6, balancing semantic diversity with topical coherence. Values below 0.3 indicate excessive divergence with potentially incoherent topic-jumping between sentences, while values above 0.7 suggest repetitive or redundant content with insufficient variation. The target range of 0.3-0.6 represents healthy diversity where generated sentences explore different aspects of a topic while maintaining semantic relevance and coherent narrative flow.

---

[2]Specifically, GPT-2's implementation by the nanoGPT repository

Finally, We compute the Grammar Quality Score (GQS), a composite metric that combines rule-based grammar error detection, spelling accuracy via spellchecker (Barrus, 2020), and readability assessment using Flesch-Kincaid grade level (Flesch, 1948). GQS ranges from 0 (poor grammar) to 1 (perfect grammar), weighting grammatical correctness (50%), spelling accuracy (30%), and readability (20%). The metric identifies errors across ten categories including subject-verb agreement, tense consistency, and punctuation, with severity-weighted scoring. For complete context and to understand what the ideal ranges are for all of these metrics, see Table 3.

Table 3: Ideal ranges for generation quality metrics

| Metric | Good Range | Interpretation |
|---|---|---|
| Perplexity | 15-50 | Lower is better (fluency) |
| Distinct-1 | 0.6–0.9 | Higher is better (vocabulary diversity) |
| Distinct-2 | 0.8–0.95 | Higher is better (bigram diversity) |
| GQS | 0.8–1.0 | Higher is better (grammatical quality) |
| Avg. Cosine Similarity | 0.3–0.6 | Moderate values best (semantic diversity) |

## F.3 LISTOPS

We perform experiment on nested math operations on lists of integers, which are a version of ListOps (Nangia and Bowman, 2018). Our ListOps setting consists of three functions: maximum, median and sum modulo 20. Our inputs range from 0 to 19.

## F.4 SHAKESPEARE

As training data we used the full Shakespeare training set, tokenized such that each character constitutes a single token. Models were evaluated across the same held out validation subset. Across all models, we used a context window of 256 tokens, dropout of 10%, the AdamW($\beta_1$=0.9, $\beta_2$=0.99), and 40k maximum update iterations using a minibatch size of 64. We varied model sizes by sweeping over embedding dimensions $(32, 64, 128, 256, 380, 512, 768)$ and the number of attention heads $(1, 2, 8)$. For the recurrent models, we additionally varied the number of layers across $(3, 6, 8)$, though this does not affect the parameter count.

We found the recurrent models to be quite sensitive to choices of learning rate and learning rate schedules. Hence, we explored several different maximum learning rates $(1e-3, 7.5e-4, 3e-4, 1e-4)$, schedules (cosine, exponential), and minimum learning rates ($10\times$, $20\times$, and $100\times$ smaller than the max learning rate). LR warm-up was 100 updates for all experiments.

In Figure 4 we emphasize the best losses we were able to achieve for each model size. In addition, we capture the model's sensitivity to hyperparams by showing the top 50% performing models across all hyperparameters.

### F.4.1 BEST MODEL CONFIGURATIONS AND METRICS

Table 4 shows the best model configuration for baseline GPTs and our model NRGPT with the respective generation quality metrics. We see that NRGPT outperforms the baseline GPT, RGPT and RGPT-parallel in terms of generation quality while it has only half of the nparams of GPT.

## F.5 OPENWEBTEXT

We perform experiments on natural language modeling using the OpenWebText corpus, which is an open-source recreation of GPT-2's WebText dataset (Radford et al., 2019). The dataset contains approximately 17GB of text with 9B tokens that came from 8 million documents. We tokenize using byte-pair encoding (BPE) with a vocabulary size of 50,257 tokens. Our training sequences are fixed-length contexts of 1024 tokens. Table 7 lists the hyperparameters and respective values/ranges used in our experiments.

Table 8 shows the best model configuration for baseline GPT and RGPT-parallel and our model NRGPT with the respective generation quality metrics. We see that the generation quality of NRGPT

Table 4: Best model configurations and quality metrics for Shakespeare. Note abbreviations: RGPT-parallel → RGPT-P, no of parameters → n_param, grammar quality score → gqs, average pariwise cosine similarity → apcs, distinct-1 → d-1 and distinct-2 → d-2.

| Model | Configuration | | | | | | Metrics | | | | |
|---|---|---|---|---|---|---|---|---|---|---|---|
| | lr | min_lr | n_layer | n_head | n_embed | n_params | perplexity | gqs | apcs | d-1 | d-2 |
| GPT | 5e-3 | 5e-4 | 8 | 4 | 64 | 0.4M | 294 | 0.913 | 0.198 | 0.751 | 0.978 |
| RGPT | 1e-3 | 1e-4 | 8 | 1 | 256 | 0.8M | 476 | 0.896 | 0.164 | 0.794 | 0.995 |
| RGPT-P | 2e-3 | 2e-4 | 8 | 1 | 128 | **0.2M** | 410 | 0.888 | 0.190 | 0.838 | **1** |
| NRGPT_H_FF1 | 3e-4 | 3e-5 | 6 | 2 | 512 | 2M | 318 | 0.901 | **0.218** | **0.797** | 0.995 |
| NRGPT_H_FF2W | 1e-3 | 1e-4 | 8 | 1 | 128 | **0.2M** | **283** | **0.975** | 0.176 | 0.765 | 0.995 |

Table 5: Best Model Configurations and Quality Metrics (average of four generations) for OWT with similar no of parameters. Note abbreviations: no of parameters → n_param, best train loss → t_loss, best validation loss → v_loss, grammar quality score → gqs, average pariwise cosine similarity → apcs, distinct-1 → d-1 and distinct-2 → d-2.

| Model | Configuration | | | | | | | | Metrics | | | | |
|---|---|---|---|---|---|---|---|---|---|---|---|---|---|
| | lr | min_lr | n_layer | n_head | n_embed | n_params | t_loss | v_loss | perplexity | gqs | apcs | d-1 | d-2 |
| GPT | 7e-4 | 7e-5 | 12 | 12 | 768 | 124M | 2.90 | 2.93 | 78 | 0.947 | 0.274 | 0.638 | **0.963** |
| GPT_Rec_Parallel | 6e-4 | 6e-5 | 6 | 12 | 1740 | 126M | 3.07 | 3.08 | 74 | **0.950** | 0.275 | 0.613 | 0.962 |
| NRGPT_H_FF2W | 3e-5 | 3e-6 | 6 | 12 | 1536 | 128M | 3.29 | 3.30 | 105 | 0.946 | 0.306 | **0.650** | 0.960 |

is very competitive with GPT and RGPT-parallel while it contains around 34M less parameters than GPT. Figure 5 shows an example of generated text by GPT, RGPT-parallel and NRGPT.

We report the best OWT training configurations for the metrics reported in the main paper in Table 2. We additionally test models at a more comparable 125M parameter scale in Table 5.

### F.5.1 MMLU

Perplexity and simple generation metrics are insufficient to understand the performance of language models on downstream tasks. Thus, we additionally characterize the equal-parameter models of Table 5 on the MMLU dataset (Hendrycks et al., 2020) using five-shot generation in Table 6. Remarkably, we find that *NRGPT outperforms both GPT and RecGPT at constant parameter count* on the average MMLU score.

As a caveat, we note that transformers at the ~125M scale generally perform poorly on these downstream tasks. For example, the original MMLU paper (Hendrycks et al., 2020) shows that GPT-3-small (2.7 billion params) and GPT-3-medium (6.7 billion params) (Brown et al., 2020) achieve a 25% success rate — the exact same as random guessing. A large GPT-2 model (1.5 billion params) (Radford et al., 2018) achieves a slightly better 32.4% accuracy, but this is still only 7% higher than random prediction. Our NRGPT model at 125M parameters (an order of magnitude smaller than the MMLU results of GPT-2) achieves a 29.3% accuracy, which is the best performance per parameter count on this task that we are aware of.

We believe that our results in Table 6 should be taken with a grain of salt, since we are testing small models and comparing between low accuracies. However, we also believe that these hint at NRGPT's advantage at greater scales.

Table 6: MMLU performance comparison using five-shot generation. The models tested in this work are of insufficient size and scale to achieve good performance on all of MMLU, so we report results by subject. Best are bold.

| Model | Astronomy | Science | CS | Economics | Medicine | Math | Global Facts | Average |
|---|---|---|---|---|---|---|---|---|
| NRGPT (128M) | 35.5 | **32.1** | **31.5** | 30.0 | 30.5 | **29.2** | **19.0** | **29.3** |
| RGPT (126M) | 28.3 | 27.1 | 28.0 | **32.1** | 25.2 | 24.1 | 17.0 | 26.0 |
| GPT (124M) | **36.8** | 29.5 | 26.2 | 26.7 | **31.2** | 27.3 | 18.0 | 28.0 |

Table 7: OWT Hyperparameters and range of values.

| Hyperparameter | Used Values |
|---|---|
| batch_size | 12 |
| block_size | 1024 |
| n_layer | [3, 6, 9, 12, 24] |
| n_head | [1, 2, 4, 6, 12, 16] |
| n_embed | [768, 1020, 1536] |
| learning_rate, lr | [1e-3 - 1e-5] |
| min_lr | [lr/10 - lr] |
| beta1 | 0.9 |
| beta2 | 0.99 |
| weight_decay | [1e-1, 1e-2] |
| gradient_accumulation_steps | 40 |
| eval_interval | 1,000 |
| eval_iters | 200 |
| max_iters | 100000 |
| warmup_iters | [100, 2000] |
| dropout | 0.0 |

Table 8: Best Model Configurations and Quality Metrics for OWT. Note abbreviations: RGPT-parallel $\rightarrow$ RGPT-P, no of parameters $\rightarrow$ n_param, grammar quality score $\rightarrow$ gqs, average pariwise cosine similarity $\rightarrow$ apcs, distinct-1 $\rightarrow$ d-1 and distinct-2 $\rightarrow$ d-2.

| Model | Configuration | | | | | | Metrics | | | | |
|---|---|---|---|---|---|---|---|---|---|---|---|
| | lr | min_lr | n_layer | n_head | n_embed | n_params | perplexity | gqs | apcs | d-1 | d-2 |
| GPT | 7e-4 | 7e-5 | 12 | 12 | 768 | 124M | **75** | **0.978** | 0.306 | 0.619 | 0.965 |
| RGPT-P | 6e-4 | 4e-4 | 12 | 12 | 768 | **85M** | 99 | 0.976 | **0.336** | 0.615 | 0.975 |
| NRGPT | 1e-4 | 7e-5 | 12 | 12 | 768 | 90M | 104 | 0.966 | 0.306 | **0.674** | **0.984** |

## F.6 UNCONSTRAINED PROJECTION MATRIX

We wonder how much it is necesary to constrain the inference rate matrix $\eta$ such that NRGPT continues to exhibit convergence, as shown in Figure 6. To test this, we repeat the experimental setting of Figure 2, training a 100-layer NRGPT model on ListOps that achieves 100% total accuracy. We then take 64 tokens from a validation set and minimize their energy. Remarkably, despite using a $\eta$ that is completely unconstrained (and therefore with no guarantee of energy minimization), we observe that the energy of tokens eventually stop fluctuating and converge to a stable state.

Though this observation is empirical and is not guaranteed to be the case should we have trained a model for a much longer time, it is still an interesting observation that suggests that NRGPT's convergent properties may be encouraged by the objective function, freeing us from needing to constrain the inference rate matrix $\eta$.

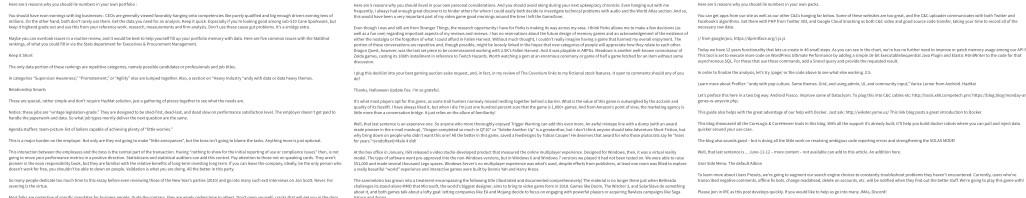

Figure 5: Best Generation Examples from GPT (left column), RGPT-parallel (middle column) and NRGPT (right column).

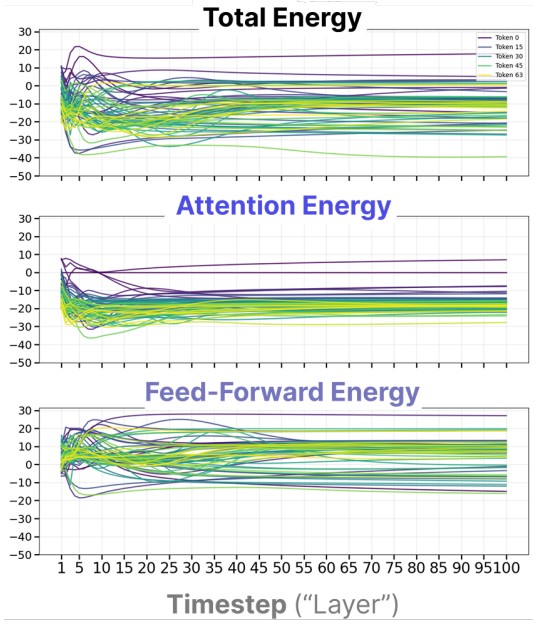

Figure 6: **NRGPT's inference rate matrix $\eta$ does not need to be constrained to induce convergent dynamics**. Shown is a 100-layer NRGPT model that achieves 100% accuracy on ListOps. Shown are 64 tokens from a validation set passed to NRGPT, where token dynamics are shown to stabilize and converge without any constraint on $\eta$.

