# OpenReview forum: "NRGPT: An Energy-based Alternative for GPT"
_ICLR.cc/2026/Conference — ICLR 2026 Poster_

### Official Review · Reviewer_d1Ma · 2025-10-29

**Soundness:** 4
**Presentation:** 4
**Contribution:** 3
**Rating:** 8
**Confidence:** 3

**Summary:**

The paper introduces an interesting and novel approach to generative modelling that integrates energy-based methods with standard GPT like methods.

**Strengths:**

Work is novel and interesting.
Modifies standard GPT methods with energy-based methods.
The work complements previous studies suggesting that transformers perform GD during inference. Unlike past work, in the proposed model inference is explicitly a gradient-based dynamics, while still maintaining an architecture very similar to GPT.

Extensive experiments show results comparable to standard GPT implementations but requiring fewer parameters.

**Weaknesses:**

Has a drawback similar to diffusion models in that inference is an iterative process and thus takes time and incurs a heavy computational burden. As the paper mentions, a T-step gradient descent can be mapped to a T-layer GPT transformer architecture, but the computational burden remains if the required number of steps T is large.

**Questions:**

The proposed method has similarities to diffusion models. Can it be used to enhance diffusion models (e,g, to generative images instead of applying it to LLMs, using a GAN discriminator to provide a score)?

Can proposition 2.1 be strengthened to provide a weaker constraint on the required inference rate matrix? (i.e. are there other inference rate matrices where the energy decreases?) There is some discussion of this, but I am mainly wondering if there are proofs where the inference rate is something that is more useful for the problem at hand.

Not necessarily a weakness, but the issue of asymptotic stability raises the question of short-term stability. Does this affect the operation and training of the network? How long does one have to wait to reach quasi-stability where things have settled down enough for reliable inference?

What is it about the EBM approach that results in needing fewer parameters than equivalent GPT models (table 2)?

---

> ### Author Response · Authors · 2025-11-25
>
> We thank the reviewer for their vote of confidence and interesting questions! We enjoyed responding to your questions and comments below:
>
> > has a drawback similar to diffusion models in that inference is an iterative process and thus takes time and incurs a heavy computational burden
>
> Amazingly, we find that relatively few recursions, consistent with the depth of conventional GPT models, yields good results. In other words, our experiments do not require a high number of iterations for good performance. However, it is possible that more iterations could help with more challenging problems. In fact, a recent ["mixture of recursions" paper](https://arxiv.org/pdf/2507.10524) has suggested something quite like this as a strategy for thinking/reasoning in LLMs.
>
> > The proposed method has similarities to diffusion models. Can it be used to enhance diffusion models?
>
> Very insightful point! Indeed, our NRGPT model was inspired by the Energy Transformer (ET) model which was applied to images (in a patch-completion manner, much like the [Masked AutoEncoder (MAE)](https://arxiv.org/abs/2111.06377) approach.).
> With small modifications, one can also use NRGPT as a purely generative model. This is because there is a deep mathematical connection to diffusion models, where minimizing energy is equivalent to maximizing the log-likelihood approximated by score-based models (see [Song et al 2021](https://arxiv.org/pdf/2011.13456)). The major difference of NRGPT with diffusion models is that NRGPT models $\log P$ explicitly, while in diffusion models the $\log P$ is modeled implicitly (i.e., there is no explicit scalar energy or log-likelihood) --- diffusion models mysteriously encode the energy gradient into the opaque weights of the model.
>
> > Can proposition 2.1 be strengthened to provide a weaker constraint on the required inference rate matrix?
>
> Great question. If we could ignore the projection $P\_A$, as long as the real part of all the eigenvalues of $\boldsymbol{\eta}$ is positive, the energy will decrease, though its imaginary part may cause the dynamics to spiral a bit. However, since $P\_A$ removes a 1D subspace spanned by $x\_A-\mu\_A$ for each token $A$, it is unclear whether on the data manifold a more relaxed condition could be found. Yet, amazingly, the LayerNorm may ensure that *almost all $\boldsymbol{\eta}$ lead to convergence*. This is because LayerNorm together with Lipschitz activation will lead to a bounded energy, which cannot explode. Therefore, unless the dynamics is exactly energy conserving (e.g. Hamiltonian), the energy will converge to some value. This doesn't guarantee decreasing energy, but it does suggest that convergence is the typical behavior for our model. The derivation of the boundedness and some corrections to the proof of 2.1 can be found in appendix E.
>
> > the issue of asymptotic stability raises the question of short-term stability. Does this affect the operation and training of the network? How long does one have to wait to reach quasi-stability where things have settled down enough for reliable inference?
>
> Regarding whether *short-term stability is sufficient for good performance* of the network, the answer is a clear **yes**. Our experiments evaluate models that are trained for a fixed number of steps where token dynamics have not completely settled --- asymptotic stability is not necessary for good performance!
> Regarding how long we have to wait to reach quasi-stability, we believe this answer is quantifiable in terms of the real and imaginary parts of the eigenvalues of the Hessian of the energy. We are happy to elaborate and discuss further with the reviewer if they are interested.
>
> > What is it about the EBM approach that results in needing fewer parameters than equivalent GPT models (table 2)?
>
> EBM approaches are not necessarily more parameter efficient than equivalent GPT models. It all depends on how the energy is parameterized. For instance, in the [Energy Based Transformers](https://arxiv.org/abs/2507.02092) work, the energy is parameterized by a standard transformer and thus the parameter count is approx. equivalent. In this work, we follow the example set by the [Energy Transformer](https://arxiv.org/abs/2302.07253) to parameterize the transformer block itself as an energy. This has an interesting effect, where the value matrix of attention shares weights with query and key matrices, and the second weight matrix $\boldsymbol{W}^2$ of the MLP is shared with its first layer weights $\boldsymbol{W}^1$.
>
> We regain independent degrees of freedom using our inference rate matrix $\boldsymbol{\eta}$. You can see this in our clarified eqs. (13) and (15) (thanks to @JUki for suggesting this clarification!)
>
> ----
>
> We once again thank the reviewer for their time and interest in our submission. We hope the above responses clarify your questions, and if they have improved your view on the quality of our submission we are always grateful for an increased score 🤗. Thank you again!

---

### Official Review · Reviewer_JUki · 2025-11-02

**Soundness:** 2
**Presentation:** 1
**Contribution:** 2
**Rating:** 2
**Confidence:** 2

**Summary:**

This paper proposes an energy-based framework for next-token prediction by making key edits to the Transformer architecture to achieve a recurrent block. This model is learned via a differentiable sampling process where parameters are learned during sampling. The energy function mimics the Transformer via a parallel block with an attention-based component and a feed-forward based component, as in parallel transformer designs. Tokens are transformed into the next via exploring the energy landscape. The block is repeatedly applied to mimic layer depth in a recurrent fashion. Results on simple tasks like domain-specific language modeling and list ops show performance comparable to a layer-recurrent Transformer.

**Strengths:**

1. This work extends prior work on energy based modeling and the Transformer archtecture to the new and popular setting of next token prediction, advancing from prior work largely focusing on masked infilling based modeling.
2. The experiments testing NR-GPT on ListOPs and Shakespeare data show competitive results with a recurrent version of a transformer model, indicating that this different learning paradigm has promise and can perform competitively with a modficiation on a dominant paradigm.

**Weaknesses:**

1. It is unclear how much novelty is presented in this work versus the immediate prior work of Energy Transformer. For example, the parallel block structure for EBM was proposed in the prior work, but not attributed directly in this work. Additionally, while proper attribution was given for the attention energy, this architecture and energy formulation are also directly from the prior work, constituting a major part of the current paper's claimed contributions.
2. The motivation for this avenue of exploring EBMs and Transformers is lacking in the current work - while I believe that studying new avenues for language modeling are important, it is not clear from this work what precise benefit EBMs may give over current language modeling, even if they are "not there yet."
3. Key details on how sampling actually occurs in this model are not present. How does the arcitecture embed tokens into the network? And more importantly, how are samples actually decoded into tokens?

**Questions:**

1. How should the original: and energy: expressions be interpreted in equations 13 and 15?
2. While it is clearly not the point of this work to outperform current language models given the exploratory and new nature of this work, I am still curious to know what are the run-time, memory, FLOP/compute use differences between this model and normal GPT?

Notes:
1. Cutoff on line 323, sentence not finished, “For”.
2. Cutoff on line 353, “To reach”, sentence not finished
3. Citations for Qwen, Llama, and RMSNorm are missing on line 245.

---

> ### Author Response · Authors · 2025-11-25
>
> We thank the reviewer for their honest and thoughtful review. You have raised good questions, especially around the story-telling and contextualization of this paper. We hope that you will forgive our lengthy response, which we hope adequately addresses your concerns with our paper.
>
> > It is unclear how much novelty is presented in this work versus the immediate prior work of Energy Transformer.
>
> The reviewer is correct in noticing a strong similarity of NRGPT to the Energy Transformer (ET). Indeed, much of the inspiration for NRGPT's design came from that prior work. However, we emphasize that NRGPT is *fundamentally different* from ET in that it tackles the key challenge of *causal generation*, a paradigm that is not compatible with the original ET's bidirectional attention design (where attention signal enables past tokens to attend to future tokens). A simple causal mask on ET breaks the guarantee of monotonic energy minimization.
>
> The key innovation of NRGPT is to model the sequence as a collection of token-wise energies $E\_A$ rather than a global sequence energy $E = \sum_A E\_A$ (where $A = 1, \ldots, N$ indexes the sequence length). We discover that *tokens still converge* even when minimizing all individual energies simultaneously, and we argue that this generalizes the ideas of ET to allow tokens to explore a meaningful energy landscape during token generation.
>
> We also theoretically and empirically study the **projection matrix** that is present in all standard transformers but that is *noticeably absent* in ET's attention formulation.
> We interpret this matrix as an "inference rate" matrix $\boldsymbol{\eta}$ used for gradient descent, where under certain conditions it guarantees token convergence. We are excited to see that including the projection matrix and modifying the FF energy makes the NRGPT block *as expressive* as a recurrent, standard GPT block, that it can compete with small GPT models on causal language modeling tasks up to the size of OWT. These conclusions were not evident in the original ET paper.
>
> In summary, NRGPT is distinct from ET because:
>
> 1. **NRGPT performs causal language modeling by minimizing a per-token energy**.
> ET was restricted to strict energy minimization of an entire sequence, a paradigm incompatible with the parallel, autoregressive language modeling of GPT-style transformers.
> 2. **NRGPT uses learnable inference rate matrices** $\boldsymbol{\eta}$ during token prediction. Meanwhile, ET was restricted to a fixed, scalar gradient descent step which did not allow additional exploration of the energy landscape.
> 3. **NRGPT explores alternative energy-replacements for the feed-forward (FF) MLP module**. ET used a single-layer Hopfield Network with energy $G(\boldsymbol{\xi} \mathbf{g}_A)$, which results in the weights of the two layers to be $\boldsymbol{\xi}$ and $\boldsymbol{\xi}^T$. In NRGPT, we explore a more general form $E^{\mathrm{FF}}$ for the feed-forward module and find improved results on this causal language modeling task.
>
> We have clarified these comments in App. B of our updated submission, and we thank the reviewer for bringing this confusion to our attention.
>
> > Key details on how sampling actually occurs in this model are not present. How does the architecture embed tokens into the network? And more importantly, how are samples actually decoded into tokens?
>
> We apologize for the omission of these details. We have updated our submission (App. F) with the details below:
>
> NRGPT is a drop-in replacement for the standard GPT block, where tokens and positions are embedded using learnable embedding matrices following the precedent of GPT-2 (specifically, the implementation by the [nanoGPT repository](https://github.com/karpathy/nanoGPT)). ``Sampling'' is a *complete forward pass through the transformer*. This is easiest to explain for when $\\eta =1$ where the forward pass is explicit gradient descent on the energy $E_A$ of token index $A$. Here, gradient descent provides a fast, differentiable way to move initial points to locations which have lower energy (higher likelihood).
> Each new token starts from an initial position $x\_A(t=0)$. The forward pass does GD, evolving it to a point $x\_A(t=t\_f)$ which has lower energy. In this case, $E\_A$ can be interpreted as the log-transition probability. In all GPT-style models (and therefore also in NRGPT) we choose the initial point $x\_A(t=0)$ to be $x\_{A-1}$, the embedding of the previous token, but tokens could be initialized to anything in theory.
>
> In the case where $\boldsymbol{\eta} \ne \mathbf{1}$, the forward pass follows more complex dynamics than pure GD, but the model learns dynamics that convert $x\_{A-1}$ to $x\_A$. This process is what we refer to as "sampling". However, since this case is not strictly a sampling process, we are happy to change our wording choice at the suggestion of the reviewer and reserve the term "sampling" for the case of pure GD. Please let us know what you would prefer.

---

> > ### Author Response · Authors · 2025-11-25
> >
> > > It is not clear from this work what precise benefit EBMs may give over current language modeling.
> >
> > Thank you for this feedback --- we have updated our submission (App. C) with the following motivation, which we believe is an important component of the story.
> >
> > EBMs offer a paradigm for generative modeling that is inherently about *optimization*, where the model samples a new token from a transition probability described by the prior context. Conventional transformers are trained to *implicitly* learn this transition probability, but it is hidden in the architectural design. The key appeal of an EBM like NRGPT is that it **models the (unnormalized) transition probability explicitly**. This offers several advantages which we hope to explore in future work:
> >
> > 1. **Systematic exploration of the solution space**. An explicit likelihood function enables us to explore the space of solutions in LLMs using well-established methods in optimization and statistical physics, such as alternative gradient descent methods, minimum-energy paths, metastability, etc. These tools are not available when the energy is implicitly encoded in a deep architecture, and we believe that finding creative solutions may correspond to exploring different local minima of the energy during inference.
> > 2. **Variable computation via early stopping criteria**. The energy can be used as a signal to stop model computation early for easier problems, or to continue thinking longer for harder ones. We discuss this benefit at length in our response to Reviewer cL58, and it is a primary motivation of other EBMs like the [EBT](https://arxiv.org/abs/2507.02092).
> > 3. **Model alignment using energy regularizers**. Eq. (7) of our paper shows that NRGPT's architecture is the sum of two energies: a token-mixing *attention energy* and a token-wise *feed-forward energy*. Per the precedent of ET, these are chosen to be faithful to the original transformer's design. However, any scalar objective can be added to NRGPT's block, and we imagine that adding regularizer terms to bias the energy landscape toward favorable solutions (or away from undesirable ones) is a benefit unique to EBMs that is more robust to prompt injection attacks than current LLMs.
> >
> > We believe there are several additional advantages that explicit EBMs have compared to traditional LMs that we are happy to discuss further. In the meantime, we have updated our submission with a succinct summary of the content presented above, and we thank the reviewer for giving us the chance to clarify this part of our story.
> >
> > > How should the original: and energy: expressions be interpreted in equations 13 and 15?
> >
> > We apologize for the confusing notation. Eqs 13 and 15 mean that the overall structure of the energy model is essentially the same as the original transformer, where the specified operations of the "original:" update are now equivalent to the new operations in the "energy:" update. Specifically, in (13) attention in both models has the form
> > $\mathrm{AT}(\mathbf{g}) = \boldsymbol{M} \mathbf{g}\mathrm{SM}(\mathbf{g}^\top \boldsymbol{J} \mathbf{g})$, but in the "original" GPT $\boldsymbol{M} = [\boldsymbol{W}^{P}]^\top \boldsymbol{W}^V$, but it is $\boldsymbol{M} = \alpha \boldsymbol{\eta} \boldsymbol{J}^T$ in the "energy" model.
> > In (15), the feedforward module in both models looks like $\boldsymbol{M} \sigma(\boldsymbol{W}^1 g)$.
> > In "original" GPT $\boldsymbol{M} = \boldsymbol{W}\_2$ whereas in the "energy" model we get $\boldsymbol{M} = \boldsymbol{W}^1 \boldsymbol{\eta}^\top$, using (14) for $E^{\mathrm{FF}}$.
> >
> > We have updated the notation in our submission to avoid confusion for future readers.
> >
> > > what are the run-time, memory, FLOP/compute use differences between this model and normal GPT?
> >
> > Please see our answer in the **General Rebuttal** above. NRGPT is anywhere from $1-2{\times}$ the FLOP cost of an equivalent RecGPT at constant parameters, which is a byproduct of applying the chain rule when doing the gradient computation during inference.
> >
> > > (re: typos)
> >
> > Thank you for finding the typos and missing citations in our work. We have fixed these in the updated submission.
> >
> > ---
> >
> > We thank the reviewer again for their candid feedback and helpful questions. We hope our above responses have clarified the story of our paper, and we are happy to engage further if you have remaining questions or concerns. If you have found our responses satisfactory, we would greatly appreciate an increase of score to reflect your increased confidence in our work.

---

> > > ### Comment · Reviewer_JUki · 2025-11-27
> > >
> > > Thanks for the detailed response and the updates in the pdf.
> > >
> > > 1. I feel my concerns re ET v NRGPT are largely clarified and well-addressed in the main paper.
> > > 2. As for the architectural details, I don't wish for you to have to change the language used, I was mostly looking for key embedding/decoding details like projecting into/out of the vocabulary space which has been detailed in Appendix F now.
> > > 3. The paper is meaningfully improved with competitive fixed param experiments across NRGPT/Recurrent GPT/GPT and more explanations as to the appeal and importance of EBM research.
> > >
> > > I'm raising my score to reflect increased confidence in the work, and kudos on improving the presentation and content of the paper to this extent in such a short period of time.

---

### Official Review · Reviewer_cL58 · 2025-11-03

**Soundness:** 2
**Presentation:** 4
**Contribution:** 2
**Rating:** 6
**Confidence:** 4

**Summary:**

The paper reformulates the GPT as an EBM. Instead of stacking multiple transformer layers, it uses a single recurrent module that performs iterative energy descent steps. Each forward pass updates the token representations by following the gradient of a total energy function that combines attention and feedforward energies, turning inference into motion along an energy landscape.

The authors derive energy formulations whose gradients reproduce the transformer’s attention and feedforward operations. They introduce a learnable inference rate matrix η(t), prove that energy monotonically decreases under LayerNorm or RMSNorm, and show asymptotic stability of the update dynamics.

Experiments demonstrate that NRGPT achieves performance comparable to standard and recurrent GPTs on ListOps, Shakespeare, and OpenWebText benchmarks, while requiring fewer parameters and showing greater resistance to overfitting.

Overall, the work provides a minimal and theoretically grounded link between GPT and EBM frameworks, showing that transformer inference can be interpreted as an explicit gradient-based energy minimization process without sacrificing model quality.

**Strengths:**

This paper stands out for its originality in redefining the GPT architecture through an energy-based modeling perspective. Instead of proposing a new model class, it reinterprets existing transformer mechanics as iterative energy minimization, offering a fresh theoretical lens rather than a purely architectural innovation.

Its technical quality is solid: the authors construct explicit energy functions for both attention and feedforward components, establish convergence and stability guarantees, and validate the model on multiple benchmarks. The experiments are carefully designed and demonstrate that the energy-based formulation can match GPT performance while using fewer parameters.

The clarity of presentation is strong, with concise mathematical derivations and intuitive explanations that make the energy interpretation easy to follow.

In terms of significance, the work bridges a conceptual gap between transformers and energy-based models, opening a path toward more theoretically grounded, interpretable, and potentially more efficient generative models.

**Weaknesses:**

First, the experiments are limited to small and medium-scale models (≤90M parameters) and simple datasets like ListOps and Shakespeare, making it unclear whether NRGPT remains stable and efficient at larger LLM scales. Scaling experiments on multi-billion parameter models and broader benchmarks such as MMLU or GSM8K would make the conclusions more general.

Second, the theoretical guarantees rely on a learnable inference rate matrix η(t) and specific normalization (LayerNorm/RMSNorm), but the paper lacks ablations on their robustness and training feasibility. Systematic studies of different η parameterizations and normalization variants would clarify when the energy monotonicity and stability conditions hold in practice.

Third, the paper does not analyze runtime efficiency or expressivity trade-offs. Because the model enforces weight sharing, it may have higher iteration costs or reduced fitting capacity, which should be quantified under equal FLOPs or wall-clock settings.

Finally, the paper should discuss its relationship to *Energy-Based Transformers are Scalable Learners and Thinkers* (arXiv:2507.02092). While that work focuses on scalable, multimodal reasoning with “System 2” inference, NRGPT formalizes energy descent within standard GPT. A clear comparison of their goals, formulations, and implications would strengthen the paper’s positioning and highlight its distinct contributions.

**Questions:**

1. Is the proposed energy formulation essentially unique, or could multiple energy functions induce the same transformer dynamics? Any intuition or simple example would help clarify interpretability.

2. Is one-step energy descent meant to be exactly equivalent to a transformer layer, or a principled approximation? If approximate, what governs the approximation quality (e.g., β, normalization, η structure)?

3. Beyond fixed iteration counts, have the authors explored adaptive criteria (energy change, gradient norm, or confidence signals) to decide when to stop the recurrent updates?

4. Do energy trajectories correlate with uncertainty, generation quality, or failure cases? Could energy serve as a runtime diagnostic or control signal (e.g., confidence gating, early stopping)?

5. Since the formulation emphasizes iterative refinement, can NRGPT naturally support test-time “extra thinking steps” (longer inference) without tuning? If so, are there tasks where this yields measurable gains?

---

> ### Author Response · Authors · 2025-11-25
>
> We thank the reviewer for their kind words and their constructive feedback. We have incorporated several of your suggestions into our updated paper and we hope that our answers below satisfy your outstanding questions. We have split our response into two comments to adequately address your excellent feedback.
>
> > experiments are limited to small and medium-scale models ($\leq 90$M params)... Scaling experiments on multi-billion parameter models and broader benchmarks such as MMLU or GSM8K would make the conclusions more general.
>
> We agree that scaling and broader benchmarks is an important next step for characterizing NRGPT. As you noted, transformers at the ${\sim}100$M scale of those we trained in the paper perform poorly on the downstream tasks you mentioned. For example, the original [MMLU paper](https://arxiv.org/pdf/2009.03300) (Tab. 1) shows that GPT-3-small (2.7 billion params) and GPT-3-medium (6.7 billion params) achieve a 25% success rate --- the exact same as random guessing. A large GPT-2 model (1.5 billion params) achieves a slightly better 32.4% accuracy, but this is still only 7% higher than random prediction.
>
> However, we believe your suggestion is still valuable at our small scale. We evaluated NRGPT against trained baselines on the MMLU dataset and reported the 5-shot results in Table 6. Remarkably, we find that **NRGPT outperforms both GPT and RecGPT at constant parameter count** on the average MMLU score (NRGPT gets 29.3\%, RGPT gets 26.0\%, and GPT gets 28.0\%). We are grateful to the reviewer for encouraging us to compare on this downstream task. We are hopeful that this advantage holds at greater scales, but we were not able to train larger models during the time-frame of this rebuttal.
>
> > Systematic studies of different $\boldsymbol{\eta}$ parameterizations and normalization variants would clarify when the energy monotonicity and stability conditions hold in practice
>
> Great suggestion. We have added an additional experiment in Appendix F.6, where we repeat the experiment of Figure 2 with **completely unconstrained $\boldsymbol{\eta}$**. That is, we train a 100-layer NRGPT on ListOPS, ensuring that we achieve a 100% validation accuracy, and analyze how energies evolve for each token. We observe that tokens *still converge*, indicating that NRGPT's convergent dynamics are robust to various configurations of a trained $\boldsymbol{\eta}$.
>
> > the paper does not analyze runtime efficiency or expressivity trade-offs. Because the model enforces weight sharing, it may have higher iteration costs or reduced fitting capacity, which should be quantified under equal FLOPs or wall-clock settings.
>
> We have updated our submission to include explicit comparison of NRGPT's FLOPS/layer vs. standard architectures (please also see our **General Response** above). These numbers are sufficient for comparing across architectures in our experiments because our experimental results are reported against similarly configured GPT and RecGPT where NRGPT uses the same number of "layers" as these baselines.
>
> > discuss relationship to Energy-Based Transformers are Scalable Learners and Thinkers (arXiv:2507.02092)... A clear comparison of their goals, formulations, and implications would strengthen the paper’s positioning and highlight its distinct contributions.
>
> Thank you for this suggestion. We have clarified our paper's relationship to the EBT work in our submission, and we expand on this connection below:
>
> Both EBT and NRGPT are methods that can be used to minimize an explicit energy during autoregressive generation. However, the methods differ in *how* that energy is modeled. Specifically, EBT computes energies of next tokens an output of a standard transformer
> forward pass, while NRGPT describes a parameterized energy whose gradient returns a transformer
> block. Repeatedly minimizing the energy by following this gradient resembles a complete transformer forward pass. We discuss more differences between NRGPT and EBT in App. B of our updated submission
>
> > Is the proposed energy formulation essentially unique, or could multiple energy functions induce the same transformer dynamics?
>
> This is an excellent question. While our energy formulation will have dynamics unique to its parameterization, it is possible for other energies to exhibit similar dynamics since *the objective of all these energies is the same*: predict the next token. For instance, the text around eq. (24) in our submission considers two different formulations of the energy representing the `FF` sub-block. The two methods will inherently have different updates and therefore dynamics, but they are both trying to model token prediction. Similarly, we could modify the attention energy in eq. (11) which will change the dynamics, but the new dynamics will have the same objective as what we have proposed in this paper. Your intuition to explore alternative energy designs is good, and there is a lot of room for designing of these formal energies.

---

> > ### Author Response · Authors · 2025-11-25
> >
> > > Is one-step energy descent meant to be exactly equivalent to a transformer layer, or a principled approximation? If approximate, what governs the approximation quality (e.g., $\beta$, normalization, $\eta$ structure)?
> >
> > A principled approximation! Indeed, when taking the gradient of our energy, the chain rule pops weights from inside non-linearities outside them, forcing us to share weights during updates --- a restriction that standard transformers do not share.
> >
> > We believe that approximation quality is best tested via downstream tasks. Consider the following: our transformer block outputs a true gradient that points in the direction of minimizing energy, whereas a standard transformer block is almost guaranteed to not be a proper gradient. Thus, our goal is to derive the transformer's update in a principled manner.
> >
> > The performance of our designed energy function is absolutely affected by the quantities you suggested, and tuning hyper-parameters like $\beta$ and projection $\eta$ are important considerations for scaling our method.
> >
> > > Beyond fixed iteration counts, have the authors explored adaptive criteria (energy change, gradient norm, or confidence signals) to decide when to stop the recurrent updates?
> >
> > These adaptive stopping criteria are an incredible advantage for EBMs over traditional methods. Interestingly, if you can guarantee that all tokens $B<A$ are fixed in NRGPT's energy, then the energy of token $A$ is guaranteed to decrease, and early stopping based on energy values and gradient norms (as you suggested) would just work. However, this constraint of a fixed context has a strong *disadvantage*, as doing so would discard the transformer's unique advantage of full parallelism across tokens. For the causal energy formulation with parallel token evolution studied in this work, we are actually not guaranteed to monotonically decrease the energy and energy deltas and gradient norms would be unreliable (see fig. 2 for token-wise energy trajectories). On the other hand, designing a dedicated network to predict confidence signals for early stopping is an excellent idea that would work with our framework, but we did not consider that in the scope of our submission.
> >
> > > Do energy trajectories correlate with uncertainty, generation quality, or failure cases? Could energy serve as a runtime diagnostic or control signal (e.g., confidence gating, early stopping)?
> >
> > The energy of a model is a proxy for the *log-likelihood* of a data point under the model's learned energy, though energy is generally defined to be a relative metric that has meaning only when considering the energies of other data points. In this sense, a token's energy trajectory shows you whether a token is growing closer to or farther from the learned data manifold over time, and in theory you should be able to use this to do many kinds of runtime diagnostics on "intermediate generation quality" (but you will have to be careful of e.g., saddle points where gradients are small but there may be much better minima beyond). These are ideas are readily in the scope of a follow-up work.
> >
> > See our answer above re: your question on runtime control signals like early stopping..
> >
> > > Since the formulation emphasizes iterative refinement, can NRGPT naturally support test-time “extra thinking steps” (longer inference) without tuning? If so, are there tasks where this yields measurable gains?
> >
> > Another very meaningful question. Indeed, in the context of only predicting a single token (rather than an entire sequence in parallel), we are iteratively improving the token's embedding, until at some point the convergence guarantees are met and there is no purpose to "thinking longer". This iterative improvement needs no fine-tuning and it is baked into the model design. Unfortunately, it is very expensive and difficult to train the convergence point of NRGPT using backpropagation, since this requires iterating gradients through many, many unrolled computation steps. As such we have not tested measurable task performance on this, but it is a very meaningful direction for future work.
> >
> > ---
> >
> > We thank the reviewer again for their feedback and their time spent reviewing our paper. If our responses above and the updates we have made to the paper have addressed your initial concerns, we would greatly appreciate an increase in score to reflect your increased confidence in our work 🙏.

---

### Official Review · Reviewer_xLun · 2025-11-04

**Soundness:** 3
**Presentation:** 3
**Contribution:** 3
**Rating:** 6
**Confidence:** 2

**Summary:**

This paper presents eNeRgy-GPT (NRGPT) a framework for autoregressive language modeling that converts a Generative Pre-trained Transformer (GPT) into an energy-based model. NRGPT inference is conceptualized as an exploration on an energy landscape, where each layer becomes a gradient descent step on an energy function comprising attention and feedforward components. The authors establish theoretical conditions under which this process guarantees energy descent.
On three datasets, nested math operations on lists of integers (ListOps), character-level modeling (Shakespeare), and standard language modeling (OpenWebText) NRGPT achieves comparable performance with GPT baselines across different parameter sizes. Notably, the model exhibits greater resistance to overfitting compared to standard transformers.

**Strengths:**

- The authors unify the well-known GPT architecture with Energy-based models, providing explicit energy functions for both attention and feedforward components. This theoretical contribution is novel and advances recent work exploring these connections.

- The paper is well-written and easy to follow.

- While the experiments on ListOps, Shakespeare, and Open Web Text cover a narrow range of tasks, they are nonetheless informative for drawing meaningful comparisons between models.

**Weaknesses:**

- While the theoretical connection between energy-based models and transformers is interesting, the paper does not clearly state why this direction is practically valuable or how it could lead to meaningful advances beyond conceptual unification.
- While NRGPT achieves performance comparable to a recurrent GPT, this comparison has limited practical relevance. The meaningful benchmark remains the standard GPT architecture.
- NRGPT requires weight sharing across layers (recurrent steps), which posits a constraint in terms of architectural flexibility.

**Questions:**

- Why not comparing NRGPT with recurrent GPT and GPT at approximately similar number of parameters?
- When comparing models in Open Web Text (Section 3.4), are you using the same tokenizer across all models?
- How do GPT/Rec-GPT differ from NRGPT in terms of computational cost. How much does the need of the inference rate matrix affect FLOPs?


Comments:

l114: This is also the premise Energy-Based Models (EBM)

l121: the final point match real datapoints

l322: TO evaluate the

l323 ends abruptly

---

> ### Author Response · Authors · 2025-11-25
>
> We thank the reviewer for the time and thoughtfulness that they put into their review. We believe we your feedback has improve our submission, and hope the responses below address any outstanding concerns you may have.
>
> ---
>
> > the paper does not clearly state why this direction is practically valuable
>
> This is great feedback. Indeed, many EBM researchers (including ourselves) chase explicit energy models for its theoretical elegance, often at the cost of practical performance. In this work, we showed that *a GPT block with explicit energy is no less expressive than a standard GPT block* across a range of language modeling tasks, which we believe opens the door for improved performance on explicit energy models downstream.
>
> Practically, our framework offers a few benefits over existing GPT blocks that are exciting directions for future work:
>
> 1. NRGPT admits an interpretation of the forward pass as both a *minimization* and an *exploration* of its learned energy landscape. This allows us to consider, in addition to the learned projected gradient descent of this work, additional sampling techniques like ADAM at prediction time.
> 2. Our model exhibits *convergence* over a sequence of tokens. This enables training via e.g., fixed-point convergence methods or other energy-based contrastive methods.
>
> We provide more motivation in our updated PDF (App. C) and in our response to reviewer JUki.
>
> > While NRGPT achieves performance comparable to a recurrent GPT, this comparison has limited practical relevance. The meaningful benchmark remains the standard GPT architecture.
>
> We agree that the benchmark remains the full GPT, which we include as a baseline in all experiments. We compare against recurrent GPT due to its structural similarity to NRGPT, making it suitable as a direct ablation against the modeling performance of our energy block + inference rate matrix $\boldsymbol{\eta}$. In our updated results, we find that NRGPT outperforms a standard RecGPT at constant parameters on downstream tasks like MMLU (Table 6), while exhibiting comparable performance on OWT, Shakespeare, and ListOps tasks.
>
> > NRGPT requires weight sharing across layers (recurrent steps), which posits a constraint in terms of architectural flexibility.
>
> Correct, weight sharing across layers is required for our recurrent optimization process. This is an architectural constraint, but we point out that weight sharing across layers is receiving renewed interest in [mixture-of-experts (MoE)-type universal transformers](https://dl.acm.org/doi/10.5555/3737916.3738813) because of the computational benefit they provide during training/inference.
> Additionally, multiple works have observed significant weight correlations across layers of LLM (e.g. [ShortGPT](https://arxiv.org/abs/2403.03853)), indicating that unconstrained parameters seem to learn some form of recurrence.
>
> In NRGPT we can recover flexibility in other forms like learnable inference rates $\boldsymbol{\eta}$ per layer and other means of exploring the explicit energy landscape that we plan to explore further in future work.
>
> > Why not comparing NRGPT with RecGPT and GPT at approximately similar number of parameters?
>
> Thank you for this suggestion! We have retrained both NRGPT and RecGPT to match the parameters of GPT and report the new results in Tables 5 and 6 of the appendix. At 125M, the losses and perplexity of GPT and RecGPT are better than NRGPT, but we find that NRGPT outperforms both these models on downstream tasks like 5-shot MMLU. We keep the original table with parameter counts ranging from 85M-125M in the main paper (Table 2) because, in those experiments, we fix the different models at a constant *width* --- i.e., we chose $D = 786$ to match the default setting of GPTs. The full details for that experiment are shown in Table 7.
>
> > are you using the same tokenizer across all models?
>
> Yes! OWT experiments use the same tokenizer across all experiments. We have clarified this and other architectural details in Appendix F.1 of our updated submission.
>
> > How do GPT/Rec-GPT differ from NRGPT in terms of computational cost. How much does the need of the inference rate matrix affect FLOPs?
>
> Re: computational cost, we refer the reviewer to the **General Response** above where NRGPT is at most 2x slower than GPT. The inference rate matrix adds no overhead compared to the transformer: i.e., if we let $D$ be the size of each token's embedding, then at its most expensive the inference rate matrix is a $D \times D$, equal-FLOPs replacement for the `proj` matrix in standard attention. The inference rate can also be a single scalar with only one FLOP per update step.
>
> > (re: typos)
>
> Thank you for identifying these! We have addressed these in our updated PDF.
>
> ---
>
> Once again, we are very grateful for your feedback. If you found that our responses above satisfied your initial concerns, we would greatly appreciate an increase in score to reflect your confidence in our submission 🙏.

---

### Author Response · Authors · 2025-11-25
**General Response**

We thank the reviewers for their thoughtful feedback and questions. We have addressed the concerns in our updated paper (changes are emphasized in **blue** of our updated PDF), and we respond to individual reviewer comments below.

We are encouraged that the reviewers agree that our work is:

1. **Well-written and clearly presented** (xLun, cL58)
2. **A novel theoretical contribution uniting GPT and EBMs** (xLun, cL58, d1Ma)
3. **Grounded in experiments showcasing NRGPT's promise across multiple datasets** (JUki, cL58, d1Ma).

We also greatly appreciate the constructive feedback shared by several reviewers, which we believe has improved the story around our paper. Specifically:

- **Extend evaluations to downstream benchmarks e.g., MMLU** (cL58). We add a new experimental result in Table 6 of our paper where we show that *NRGPT outperforms both GPT and recurrent GPT at constant parameter count* on average MMLU score. We believe this provides a strong signal that NRGPT can scale beyond the experiments reported in this work.
- **Clarify NRGPT's relationship to prior work** (cL58, JUki). There are two works mentioned, the "Energy Based Transformers are Scalable Learners and Thinkers" (arXiv:2507.02092) and "Energy Transformer" (arxiv:2302.07253). We have updated the our paper with extended discussion on both of these.
- **Clarify benefits of an explicit EBM language model** (xLun, JUki). We improved our paper to provide better motivation in the intro and appendix, and we believe many of the reviewers' questions hint at the inherent advantages of explicit EBMs.
- **Include computational efficiency comparisons** (xLun, cL58, JUki, d1Ma). We have included these comparisons in our updated paper's supplementary, and summarize the results in this general response below.

We take this opportunity to reiterate a few key points about our paper and summarize our response to the shared comments raised by the reviewers above.

## Key points
1. **Paradigm:** In NRGPT, the inference process is formally exploring an energy landscape that we believe is implicitly encoded in the forward pass of a standard transformer.
2. **Contrast with Energy Transformer:** NRGPT extends the Energy Transformer (ET) to language modeling by introducing:
    1. causal masking;
    2. per-token energy;
    3. learnable inference rate matrices;
    4. Alternative MLP modules;
    5. Uses standard LayerNorm and positional embedding.
3. **Relation to Diffusion models:** Diffusion models, learn to follow gradients of $ E = -\\log P$, *implicitly* encoded in the weights.
In contrast, NRGPT models $E$ *explicitly*.
4. **Benefits of Energy Based:** An explicit energy may allow for:
    1. **exploration/exploitation on energy landscape:** the inference strategy could be modified for creativity, reasoning and finding alternative solutions.
    2. **Potential robustness:** Alignment and steering of the model could be done by regularizing the energy.
    These can be done at inference time, are flexible, and could potentially be more robust than prompting.

## Regarding FLOPS
We are adding this important discussion to the paper in Appendix D and Limitations. The key points are:

- **NRGPT Attention:** For a single head, the attention block of NRGPT costs fewer FLOPS than recurrent and normal GPT, as NRGPT uses shared key-query weights $\\boldsymbol{J}$ instead of the $\\boldsymbol{W}^V$.
For multiple heads, NRGPT costs more because our current solution uses $D\\times D$ weights $\boldsymbol{J}_h$ for values, while GPT uses $D \\times (D/H)$ matrices $\\boldsymbol{W}^V$.
- **NRGPT FF:** this costs the same as an FF of the same shape in Rec-GPT and GPT.
However, we found that increasing the number of parameters and FLOPs in this module improved our performance, so the NRGPT models reported in the paper have about 2x more FLOPs than the of FF components of Rec-GPT and GPT:
    1. FF1, $E^{FF} = \\| \\sigma(\\boldsymbol{Wg})\\|^2$: In ListOps we found that FF1 competes with Rec-GPT only when the hidden dimension of $\\boldsymbol{W}$ was $8D$ instead of the standard $4D$ used in GPT.
    Though this is a constant-parameter operation, it doubles the FLOPS cost of NRGPT.
    2. FF2W, $E^{FF} = \\boldsymbol{g}^T \\boldsymbol{W^2} \\sigma(\\boldsymbol{W^1g})$, the update rule $-\\boldsymbol{\eta } \\nabla E $ leads to two terms, each being a two layer neural net. We keep the $4D$ expansion used by standard transformers and thus the same parameter count, but this leads to a ${\sim}2\times$ FLOPS in this module.
- **Memory-FLOPS tradeoff:** Even though NRGPT is more efficient when it comes to parameters, we see best performance when we increase its number of FLOPs. This is evident in the fixed-width experiments of ListOps and Shakespeare in regions where NRGPT and Rec-GPT had better test loss than GPT at the same parameter count --- these regions used wider layers and thus more FLOPs to achieve the reported performance.

---

---

### Author Response · Authors · 2025-12-02
**Rebuttal summary**

To our new AC,

Thank you leading the decision around our paper. We do not envy the amount of work you've been assigned because of the anonymity bug, and we understand the additional burden that comes from no longer being able to share the decision alongside the other reviewers.

We would like to use this comment to faithfully summarize the updates we made to our paper in response to reviewers' original reviews (changes are highlighted in blue of our updated submission). We have been grateful to our reviewers for their reasonableness and quality feedback during this discussion period, which has led to the updated scores before the reversion: **6, 6, 6, 8**.

- Reviewer xLun asked us to **compare our model against similarly sized parameter baselines**. We have done this in Table 5 of Appendix F.5.
- Reviewer cL58 asked us to **evaluate on larger-scale, downstream language tasks e.g., MMLU**. We do this and find that *NRGPT models outperform similarly sized GPT models* by a noticeable margin (Table 6 in Appendix F.5.1).
- Reviewers cl58 and JUki asked us to **clarify NRGPT's relationship to prior work** (specifically, the "Energy Transformer" and the "Energy Based Transformers"). We have written an extended discussion on their similarities/differences in Appendix B, and a summary in section 2.1 when we introduce the energy of NRGPT.
- Reviewers xLun and JUki asked us to **motivate the benefits of energy for language modeling**. We have added additional motivation to our introduction and Appendix C.
- Reviewers xLun, cL58, JUki, and d1Ma asked us to **analyze the computational efficiency** of our method (in terms of FLOPs). We have done this in Appendix D of our new submission. Conclusion: an NRGPT block uses anywhere from 1x-2x more FLOPs than a standard GPT block. This has been added to our Limitations section
- Reviewers xLun and JUki ask us to **clarify the broader architecture of NRGPT** (e.g., how tokens are embedded and sampled). We have added a complete description of the architecture in Appendix F.1.
- Reviewer cL58 asked us to **study how different choices of our "inference rate" and learning matrix $\boldsymbol{\eta}$ affect energy stability**, both in practice and in theory. We add an experiment showing that energies *still converge* with unconstrained $\boldsymbol{\eta}$ in Appendix F.6, and we theoretically characterize the importance of layernorm in Appendix E.

We believe our revised paper has addressed all the original concerns of the reviewers (please see [JUki's response](https://openreview.net/forum?id=B3Muyi2zgo&noteId=q4de3DWv8v), for example, who increased their score to 6). We hope the above summary has assisted your evaluation of our work, and we thank you again for your service 🙏.

---

### Meta-Review · Area_Chair_pmMZ · 2025-12-30

**Summary:**

This submission proposes NRGPT (eNeRgy-GPT), an energy-based reformulation of GPT-style autoregressive language modeling. The core idea is to make inference an explicit energy landscape exploration, where recurrent application of a single block corresponds to iterative (approximate) energy descent. The paper provides theoretical conditions under which the dynamics yield energy decrease and stability (notably relying on LayerNorm/RMSNorm and an inference-rate matrix), and evaluates the approach on ListOps, Shakespeare, and OpenWebText, reporting competitive performance with GPT and recurrent GPT baselines plus indications of improved resistance to overfitting. During the discussion, the authors added parameter-matched baselines, downstream evaluation (MMLU), efficiency/FLOPs analysis, clearer positioning vs closely related energy-transformer prior work, and missing architectural/sampling details.

Overall, the reviewers converged on a moderately positive assessment after the revision, with several initially serious concerns substantially alleviated during discussion. Early objections centered on fairness of evaluation, insufficient architectural clarity, unclear computational cost, and potential overlap with prior energy-based transformer formulations. These were largely addressed by adding parameter-matched baselines, downstream evaluation on MMLU, explicit FLOPs analysis, and a clearer articulation of how NRGPT differs from prior energy-based or “energy transformer” approaches in the causal, autoregressive setting. Missing or ambiguous architectural and sampling details were also filled in, resolving concerns about reproducibility and attribution.

However, a key residual concern remains around scaling and generality. Evidence is still confined to small-to-medium models and a narrow set of language modeling benchmarks, with only a limited snapshot on a reasoning-oriented task. As a result, the paper’s practical impact and robustness at realistic LLM scales are not yet convincingly demonstrated. While the conceptual framing is interesting and the revised empirical story is coherent, the claimed advantages are supported more by plausibility and early signals than by decisive large-scale validation.

Taking these points together, the reviewers’ final positions suggest that the work is technically sound and novel enough to merit serious consideration, but remains borderline due to unanswered questions about scalability and broader applicability.

**Reviewer Concerns:**

Several core concerns were substantially addressed in the revision and discussion. Reviewers asked for fairer comparisons at matched parameter counts, which the authors added (Appendix tables). Requests to test beyond the original small suite were partly met via adding MMLU (5-shot) results showing a meaningful advantage over GPT/RecGPT at similar parameter count. Multiple reviewers flagged incomplete architecture/sampling/embedding/decoding details and writing issues (cut-off sentences, missing citations); these were corrected with a more complete specification in the appendix and general cleanup. Reviewers also pressed on computational efficiency; the authors added an explicit FLOPs discussion concluding that an NRGPT block is roughly 1–2× the FLOPs of a standard GPT block, depending on configuration, and clarified that the inference-rate matrix need not add meaningful overhead in its simplest form. Finally, concerns about novelty and relation to prior “Energy Transformer” / “Energy-Based Transformers” were directly addressed with extended positioning and a clearer statement of what is new for causal generation and token-wise energies.

Some concerns remain only partially addressed. The strongest residual gap is scaling and generality: most evidence is still from small-to-medium models and limited LM-style tasks, with no demonstration at large LLM scales or on broader reasoning-heavy suites beyond the added MMLU snapshot. The paper’s practical value proposition, while improved, still relies more on plausible future benefits

**Reviewer Scores:**

Reviewer xLun initially scored 6 with reservations about practical motivation, parameter-matched comparisons, and compute cost; these were addressed via new experiments, clearer motivation, tokenizer clarification, and FLOPs analysis. I expect xLun would stay at 6.

Reviewer cL58 scored 6 and asked for scaling/downstream tasks, robustness/ablation on inference-rate and normalization, runtime efficiency, and clearer relationship to prior EBM-transformer work; the authors added MMLU, additional η experiments, FLOPs discussion, and expanded related-work comparison. I expect cL58 would stay at 6.

Reviewer JUki started at 2 due to novelty/attribution concerns and missing architectural details; after the rebuttal they explicitly raised their score to 6.

Reviewer d1Ma scored 8, noting novelty and raising thoughtful questions about diffusion connections and stability/efficiency; the authors’ responses were consistent with that positive assessment. I expect d1Ma would stay at 8.

---

### Decision · Program_Chairs · 2026-01-26

Accept (Poster)